*A Nature Portfolio journal*

# Cortical connectivity, local dynamics and stability correlates of global conscious states
Yun Zhao [1,2,3], Naotsugu Tsuchiya [4,5,6], Mario Boley[1], Vidushani Dhanawansa[1], Yueyang Liu[1], Philippa J. Karoly[7], Andria Pelentritou[8], William Woods [9], David Liley[9,10] & Levin Kuhlmann [1] ✉

Waking levels of human consciousness are known to be supported by the integrity of complex structures and processes in the brain, yet how they are exactly regulated by neurobiological mechanisms remains uncertain. Here a space-time-resolved inference-based framework is applied to estimate the neurophysiological variables of a whole-cortex model and analyze the neural mechanism correlates of global consciousness by way of a correlation analysis between behavioural and neural variable time-series. Using magnetoencephalography (MEG) data from 15 participants under Xenon-induced anesthesia, interconnected neural mass models (NMMs) were developed and time-evolving regional neurophysiological variables and inter-regional connectivity strengths were inferred from the data. Analyses revealed significant correlations between consciousness levels and inter-regional connectivity, particularly in posterior parietal, occipital, and prefrontal regions. Moreover, results support a parietal, rather than frontal, network backbone to facilitate global consciousness. Regional-level analyses further identified correlates of consciousness within the posterior parietal and occipital regions. Lastly, reductions in consciousness were linked to stabilized cortical dynamics, reflected by changes in the eigenmodes of the system. This framework provides a novel, inference-based approach to investigating consciousness, offering a time-resolved perspective on neural mechanism correlates during altered states.

Consciousness refers to the subjective awareness and perception of one's own thoughts, feelings, and the surrounding environment[1]. Global consciousness emerges when a person transitions from a state of dreamless sleep to wakefulness and remains active throughout the day, ceasing only when the individual falls asleep, loses awareness in a coma, or experiences death. Global consciousness is also altered in disorders of consciousness or during anesthesia. Many, if not all, philosophers, psychologists, and neuroscientists believe that consciousness is fundamentally a biological phenomenon[1–8]. Assuming that this belief is reasonable, one might ask " How exactly do brain processes cause conscious states and how exactly are those states realized in brain structures?"[1]. This question can be broken down into: " What exactly

are the neurobiological correlates of conscious states (NCCs), and which of those correlates are actually causally responsible for the production of consciousness?"[1].

The NCC refers to the specific systems or processes in the brain that are directly associated with conscious experience[9]. Considerable progress has been made in understanding the NCC[4,5,8]. Currently, many candidate NCCs are suggested by researchers to include brain structures and regions, neural network connectivity, and local dynamics[10–12]. Pertaining to brain structures and regions, growing evidence points to a possibility that consciousness is not supported by a single area but by a network of dispersed brain regions[13–15]. For example, global workspace theory[5,16], which is one of the

[1]Department of Data Science and Artificial Intelligence, Faculty of Information Technology, Monash University, Clayton, VIC, Australia. [2]Brain and Mind Centre, The University of Sydney, Camperdown, NSW, Australia. [3]School of Health Sciences, The University of Sydney, Camperdown, NSW, Australia. [4]School of Psychological Sciences and Turner Institute for Brain and Mental Health, Monash University, Clayton, VIC, Australia. [5]Center for Information and Neural Networks (CiNet), National Institute of Information and Communications Technology (NICT), Suita, Japan. [6]Advanced Telecommunications Research Computational Neuroscience Laboratories, Kyoto, Japan. [7]Department of Biomedical Engineering, The University of Melbourne, Melbourne, VIC, Australia. [8]Laboratoire de Recherche en Neuroimagerie (LREN), University Hospital (CHUV) and University of Lausanne (UNIL), Lausanne, Switzerland. [9]School of Health Sciences, Swinburne University of Technology, Hawthorn, VIC, Australia. [10]Department of Medicine-St Vincent's Hospital, The University of Melbourne, Parkville, VIC, Australia. ✉e-mail: levin.kuhlmann@monash.edu

leading theories of consciousness, suggests that consciousness arises from ignition and broadcast of information between various neural networks and the global neuronal workspace, such as those in the prefrontal cortex[5]. The involvement of the prefrontal cortex is a highly debated topic in contemporary research on consciousness[17–20]. A review of lesion, stimulation, and recording studies suggests that the posterior cortex, specifically the temporal, parietal, and occipital areas, is crucial for defining consciousness contents, whereas evidence for the frontal cortex's direct role in this regard is either missing or unclear[19] (but also see[21–23]).

Pertaining to neural network connectivity, researchers have proposed various models and hypotheses to explain how neural networks in the brain give rise to consciousness[24–27]. In particular, the connectivity of all default mode network regions was negatively correlated with the degree of clinical consciousness impairment[26]. Also low levels of consciousness involved decreased integration, increased segregation, and reduced global coupling of a whole-brain computational network model[27].

Local dynamics delves into the intricacies of neuronal firing patterns, revealing how the timing, rate, and synchrony of neuronal activity are integral to conscious awareness[28–30]. It also examines the specialized roles of microcircuits within specific brain regions, focusing on their unique contributions to processing and creating conscious experiences[31]. Central to this perspective is the study of neurotransmitter and synaptic dynamics, which highlights how chemical interactions at synapses influence consciousness at a local level[15,32,33].

In summary, accumulating neuroscientific evidence on the NCCs indicates that consciousness emerges from the dynamic interactions among various neural correlates, rather than being solely attributable to these correlates[34–36]. Despite such an agreement in the field, there has been no framework for researchers to "simultaneously" consider all possibly relevant NCCs and their interplays in a time-resolved manner to elucidate how they regulate consciousness as a whole. This is the gap that our study tries to fill. Specifically, the problem has been approached using computational brain models. Nonlinear computational whole-cortex models were built with regional neurophysiological variables and inter-regional connectivity as the NCCs of interest. This was achieved by augmenting our previous Neurophysiological Process Imaging (NPI) framework. NPI models dynamics of cerebral cortex at a high temporal resolution and a broad spatial scale by constructing neural mass models (NMMs)[37] in a region specific way, yet also by connecting them inter-regionally according to inter-regional connectivity derived from data. Here this new framework has been applied to magnetoencephalography (MEG) data in 15 human subjects undergoing Xenon-induced anesthesia to study the neural mechanistic basis of drug-induced reductions of consciousness in terms of local dynamics, network connectivity, and cortical stability at the same time. Xenon is known to be a putative N-methyl-D-aspartate (NMDA) receptor antagonist that effectively reduces consciousness by reducing excitation. Xenon was chosen as the anesthetic in this study to complement, but also look for similarities with, the vast majority of other anesthesia studies that predominantly focused on using gamma-aminobutyric acid (GABA) receptor agonists such as propofol or sevoflurane[19].

To gain insights into the aforementioned neural mechanistic correlates of global consciousness, model variables were estimated over time from brain imaging data at the spatiotemporal resolution afforded by MEG. Correlation analyses were conducted to examine the relationships between the level of task responsiveness (a surrogate measure of the level of consciousness) and local dynamics and network connectivity, respectively. Specifically, participants' behavioral performance was measured using an auditory continuous performance task (ACPT). The ACPT is designed to assess vigilance and responsiveness: participants hear tones of different frequencies-1 or 3 kHz-through headphones and must press one of two corresponding buttons to indicate which tone they heard. The accuracy and speed of their responses provide a quantitative measure of their alertness (or level of consciousness) as they progress through varying levels of sedation. Because it relies on sustained attention and working memory, the ACPT accuracy is sensitive to changes in consciousness and can thus be leveraged

as a reliable surrogate measure for global conscious states. By correlating our model estimates with performance on the ACPT, we aim to capture how changes in underlying neural mechanisms reflect behavioral responsiveness and, by extension, global consciousness. To elucidate a potentially critical interplay between brain networks, this study also quantified how activity in one network influences the activity in another. Moreover, dynamic cortical stability was identified as a possible global NCC measure anchored within the framework of dynamical systems theory and using neurophysiologically plausible models constrained by data.

## Results

Whole-cortex models of individuals were built, incorporating canonical NMMs, which are connected to each other based on the empirically estimated inter-regional connectivity, as shown in Fig. 1a. The neural mass model (NMM) consists of three primary neural populations: excitatory pyramidal cells (p, shown in blue in Fig. 1a), spiny stellate excitatory cells (e, depicted in green in Fig. 1a), and inhibitory interneurons (i, highlighted in red in Fig. 1a). Together, these populations form a simplified representation of a localized cortical column, mirroring the structural organization of a small region in the cerebral cortex[38]. The pyramidal cell population receives external input from a source $\mu$ (illustrated in purple in Fig. 1a), which drives activity in both excitatory and inhibitory interneurons. These interneurons, in turn, modulate the activity of the pyramidal cells through feedback mechanisms: excitatory feedback enhances activity, while inhibitory feedback suppresses it. The variable $\mu$ represents the postsynaptic membrane potential generated by neuronal activity spanning different cortical regions. The dynamic state of each neural population is characterized by its mean membrane potential, $v_n$, where $n \in \{p, e, i\}$ corresponds to the pyramidal, excitatory, or inhibitory populations, respectively. The interaction strength between two populations is defined by the parameter $\alpha_{mn}$, which denotes the average connection strength across synapses. For instance, the connectivity from the pyramidal population ($m = p$) to the excitatory population ($n = e$) is specified as $\alpha_{pe}$. This parameter aggregates several factors, including the density of synaptic connections, averaged maximum firing rate, and synaptic gain. To capture inter-regional interactions, the matrix $\mathbf{W}$ encodes the inter-regional connectivity between different NMMs, quantifying the strength of their interdependencies. The parameters of the whole-cortex model were inferred by fitting it to magnetoencephalography (MEG) source time-series data. In this process, each MEG time series was aligned with the pyramidal membrane potential of a corresponding NMM using a time-resolved inference framework.

The MEG data was collected from 15 healthy participants during wakefulness and progressively increasing sedation induced by Xenon, up to a stabilized level of 1.3 minimum alveolar concentration-awake (MAC-awake). Throughout this process, participants' responsiveness was monitored using the auditory continuous performance task (ACPT)[39] (see "Methods"). 1.3 MAC-awake represents an interesting concentration as it brings most, but not all people to a state of loss of responsiveness to verbal command. Thus it represents a reasonable concentration that brings participants in the vicinity of the boundary of consciousness/unconsciousness. Moreover, deeper levels of anesthesia where not studied for safety reasons because this data was collected in a non-hospital environment.

The ACPT presents tones of different frequencies and asks participants to respond with button presses. From the accuracy with respect to determining which tone was delivered, responsiveness (a surrogate for consciousness) levels were estimated during anesthesia delivery. Although it is well known that unresponsiveness is not the same as unconsciousness[40], for ease of exposition it is assumed that the responsiveness level (measured here as ACPT accuracy) is a monotonically increasing nonlinear function of the global consciousness level. Thus until the discussion section the words 'responsiveness level' and 'consciousness level' are treated as interchangeable.

For each participant, MEG source time series were reconstructed for each brain structure of the cerebral cortex as defined in the Automated Anatomical Labeling Atlas[41], resulting in 78 region of interest (ROI) source-

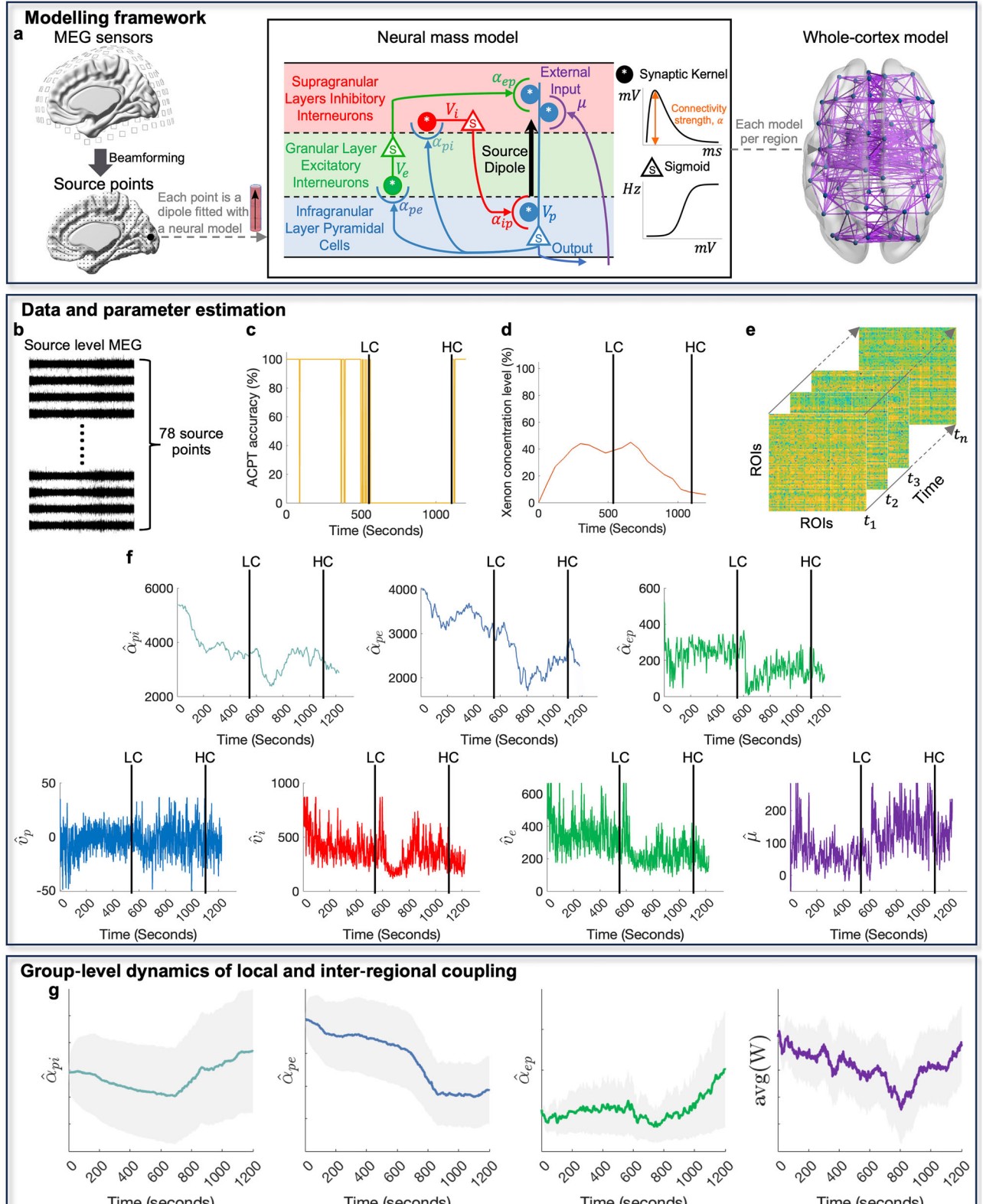

**Fig. 1 | Whole-cortex modeling and parameter estimation framework. a** The schematic of the whole-cortex model fitted to source-level MEG data is shown. Each MEG source time series is modeled using an NMM, which represents a node in the whole-cortex model. The connections between the NMMs are indicated by purple lines. **b** 78 ROI MEG source time series from an example subject during Xenon-induced anesthesia. **c, d** ACPT accuracy time series and Xenon concentration levels for the subject. LC and HC represent the estimated onset of low- and high-level consciousness, respectively. **e** Time-series estimation of the strength of inter-regional connectivity between NMMs. **f** Time-series estimation of local model states and connectivity parameters was performed for an NMM fitted to the right pre-cuneus of the subject. **g** Time courses of averaged local and inter-regional connection strength across all subjects, with the 95% confidence interval shown as gray shaded bands.

level MEG time series. As an example case, Fig. 1b-f illustrates the neurophysiological variable estimation results for an individual. Figure 1b shows the source-level time series of the cerebral cortex under Xenon-induced anesthesia. Figure 1c, d shows the ACPT accuracy and Xenon concentration levels, respectively, over the course of the experiment. As a result of the ACPT being easy, accuracy was always at 100% (i.e., the trial was correct, implying a high consciousness level), unless either the participant got the trial wrong because they were not paying attention or they just did not respond because of the consciousness-altering effects of the anesthetic (both of these cases were treated as 0% accuracy or low consciousness level). Manually identified times of transition to low consciousness level (LC) and high consciousness level (HC) are labeled in the graph by the vertical lines. Figure 1e depicts the estimated time-evolving inter-regional connectivity from the data in Fig. 1b. Figure 1f shows parameter estimates of a NMM fitted to the right precuneus (a ROI implicated previously in loss of global consciousness studies[19]). In this local NMM, the strength of excitatory connections (i.e., $\alpha_{phi}, \alpha_{pe}, \alpha_{ep}$) decreased significantly when the subject had low consciousness level. Membrane potentials ($v_i, v_e$) behaved similar to these excitatory connections, except for the pyramidal neurons ($v_p$). On the other hand, the external input ($\mu$), representing the sum across inputs from all other areas into the right precuneus, sharply increased at the onset of low consciousness level with high variation until high consciousness level was regained. Figure 1g illustrates the group level dynamics of both local and inter regional coupling under Xenon induced anesthesia across all subjects. During the reduced consciousness, the mean strengths of the pyramidal to inhibitory population connection ($\alpha_{pi}$) and the pyramidal to excitatory connection ($\alpha_{pe}$) both exhibit a clear downward shift from baseline. The averaged inter regional connectivity strength declines significantly during low-consciousness periods and then rebounds as consciousness is regained. Together, these group level results confirm that reduced levels of consciousness are accompanied by a selective weakening of local excitatory connections and by an overall reduction in long range cortical coupling.

Model fitting in our framework is performed using an analytic Kalman filter whose performance was comprehensively evaluated in another paper[37], and those results demonstrate that the filter can accurately track raw source-space MEG signals while recovering underlying model parameters with high precision. The study showed that the recovered parameters closely match their ground-truth values with errors below 6%, indicating that the fitted parameter estimates are sufficiently precise to serve as biophysically meaningful representations of the underlying cortical processes. The same modeling framework has been applied to study epileptic seizures[42] and resting state alpha rhythm[37]. In this study, we present a subject-wise RMSE boxplots in Fig. S2 showing that prediction errors remain below 10% of each signal's dynamic range across all fifteen participants, confirming that the Kalman filter effectively attenuates high-frequency noise yet faithfully preserves the slower oscillatory components that carry the core brain dynamics.

In the following analyses, the correlation between consciousness levels and inter-regional connectivity is examined, identifying brain structures and connections that are significantly associated with the level of consciousness. Next, the correlation between consciousness levels and functional networks is examined, focusing on the impact of intra- and inter-network connectivity on the level of consciousness. The relationship between local neurophysiological variable dynamics and consciousness levels is then analyzed. Finally, consciousness level changes and their association with dynamic cortical stability are investigated.

## Correlation of consciousness and inter-regional connectivity
Key brain networks associated with consciousness were identified by analyzing the temporal correlations between consciousness levels (i.e. ACPT accuracy) and centrality measures derived from the estimated inter-regional connectivity. These centrality measures assess the significance of a brain structure within the network, based on the strength of its connections to other regions through either incoming (in-degree) or outgoing (out-degree) connections. To ensure the robustness of our findings, a multiple

comparisons permutation test was used. This is a statistical method that controls for false positives when testing multiple hypotheses by randomly permuting the data, recalculating test statistics, and comparing the observed statistic to the resulting null distribution to determine significance (see "Methods"). With this approach at $\alpha = 0.05$, group-level significant "in-hub" and "out-hub"-brain structures were identified with in-degree and out-degree centrality measures that showed statistically significant correlations with consciousness levels.

The spatial distribution of in-hubs and out-hubs is illustrated in Fig. 2a, b, respectively, with the corresponding statistics provided in Table 1. Notably, prominent negative correlations were observed between consciousness levels and centrality measures in the occipital, posterior parietal, and prefrontal regions. In particular, the right middle occipital lobe, right inferior occipital lobe, left superior parietal gyrus, and left angular gyrus showed significant decrease in both in-degree and out-degree centrality measures during consciousness reduction.

We also calculated the temporal correlation between consciousness levels and the strength of each inter-regional connection. Significant connections at the group-level were identified using a multiple comparisons permutation test (see "Methods") with a significance level of $\alpha = 0.05$ on correlation coefficients across all subjects. As shown in Fig. 2c, the strength of connections within and between the parietal and occipital regions is significantly correlated with consciousness levels across some inter-regional connections. Positive correlations were observed indicating that stronger connectivity in these regions corresponds to higher levels of consciousness. Notably, four out of five connections between the parietal and occipital regions originated from parietal structures, underscoring the dominant role of the parietal regions in influencing consciousness. The other connections were within the occipital regions suggesting that the connection of internal networks in the occipital regions also substantially affects the degree of consciousness. These findings suggest that variations in consciousness levels are closely linked to connectivity strength in the occipital, posterior parietal, and prefrontal regions, with a particular preeminence of parietal over occipital regions in modulating consciousness levels.

## Correlation of consciousness and functional networks
The role of functional networks in relation to consciousness levels was explored. Seven functional networks were considered[43,44]: Auditory Network (AN), non-ventral non-anterior Default Mode Network (DMN), anterior Default Mode Network (DMNa), ventral Default Mode Network (DMNv), Executive Control Network (ECN), Sensorimotor Network (SMN), and Visual Network (VN). The specific brain structures within each network are listed in Table S1. These networks are involved in various cognitive and perceptual processes, such as categorization, attention, memory, reasoning, sensory processing, movement, and decision-making, which operate at both conscious and subconscious levels[5,45]. Understanding how these networks function during reduced consciousness is crucial for clarifying the relationship between brain activity and consciousness.

For each network, the time-varying intra-network connection strength was calculated by averaging the strength of connections within the network, and the inter-network connection strength was computed by averaging the strength of connections between functional networks. Correlation analyses between consciousness levels and both intra- and inter-network connection strengths were conducted, with the results summarized in Fig. 3 (see Table 2 for detailed statistics). The connection strength within the VN, DMNv, and DMN networks showed strong positive correlations with consciousness levels. Consciousness was particularly associated with connections from the DMN to other functional networks and from other networks to the DMNv. Significant positive correlations were also found with the VN's connections: its bidirectional links with DMN and DMNv, outward connections to SMN and AN, and inward connection from ECN. These findings suggest that changes in consciousness levels are influenced by both the internal connections within the DMN, DMNv, and VN networks and the inter-network connections linking these networks with others.

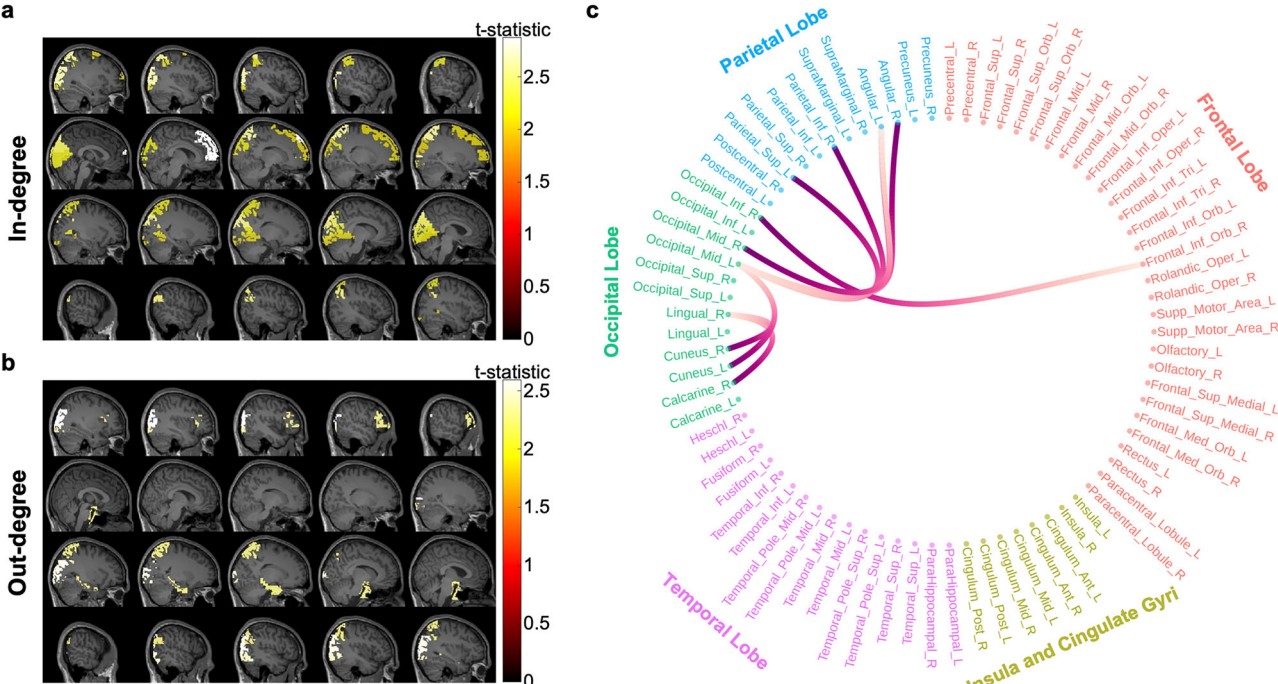

**Fig. 2 | Group-level significant hubs and inter-regional connections in relation to consciousness levels (i.e. ACPT accuracy). a** Important in-hubs associated with consciousness levels are illuminated on the template MRI images. **b** Important out-hubs associated with consciousness levels are illuminated on the template MRI images. For both **a, b**, the colorbar denotes t-statistics obtained from multiple comparisons permutation tests (see " Methods"). **c** Significant inter-regional connections most correlated with consciousness levels. The darker end represents the originating point of the connection, while the lighter end represents the termination point. Brain structures and corresponding statistical information for **a, b**, and **c** are detailed in Table 1.

## Correlation of consciousness and prefrontal and posterior parietal networks

To examine the relative contributions of the prefrontal and posterior parietal networks in modulating consciousness levels, the analysis was extended to include these networks (see Table S2 for detailed information on the specific brain structures involved)[4,19,21,46]. Correlation analyses were conducted for each network to explore the relationship between consciousness levels and both intra-network and inter-network connection strengths.

A permutation test (see " Methods") revealed significant positive correlations between consciousness levels and two connection types: the intra-network connection strength within the posterior parietal network and the inter-network connection strength from the posterior parietal to the prefrontal network (statistical details are provided in Table 3). Conversely, no significant correlations were observed between consciousness levels and the intra-network connection strength of the prefrontal network or the inter-network connection strength from the prefrontal to the posterior parietal network.

To complement our inter-regional connectivity findings and provide a functional perspective on information exchange especially since weighted symbolic mutual information (wSMI) is a well validated biomarker of consciousness levels (e.g., sensitive to distributed, time synchronized signaling across neural populations)[47,48], we computed wSMI for four networks: whole brain, posterior parietal cortex, prefrontal cortex, and the posterior parietal to prefrontal connections and correlated these measures with ACPT accuracy (i.e., consciousness levels). Figure S1 shows the time-resolved relationship between ACPT accuracy and wSMI for an example subject. Having identified that posterior parietal network and posterior parietal to prefrontal connection strength were associated with ACPT accuracy, we sought to determine if functional integration in the same regions was correlated with ACPT accuracy. Importantly, we show that (right side of Table 3) greater wSMI within the posterior parietal network, the posterior parietal to prefrontal network and across the whole brain, all correlate with ACPT accuracy.

Because wSMI quantifies the extent of functional integration, these findings suggest that subjects with more coherent, functionally integrated activity particularly in posterior parietal regions tend to have higher level of consciousness. Crucially, this biomarker converges with our earlier inter-regional connectivity results: the intra-network connection strength of the posterior parietal network and its projections to prefrontal network were significantly correlated with ACPT accuracy. In other words, both inter-regional connectivity and functional measures single out posterior parietal and its communication with prefrontal cortex as essential for modulating conscious states.

## Correlation of consciousness and regional neurophysiological variables

To gain deeper insights into local neural mechanisms, estimated regional neurophysiological variables of the model were correlated with consciousness levels. The regional variables focused on here are the parameters of the NMM fitted to each brain structure reflecting local dynamics. These include the local excitatory post-synaptic potential (EPSP) amplitudes connecting the three model populations and the external cortical input from other cortical areas. To assess their impact, we calculated the temporal correlation between consciousness levels and the NMM parameter estimates. Group-level significant regions for each variable were identified using multiple comparisons permutation tests ($\alpha = 0.05$) on correlation coefficients across subjects. These regions were then visualized on template MRI images in what we refer to as " correlation imaging".

Figure 4a shows positive correlations between the connectivity variable linking pyramidal to inhibitory populations ($\alpha_{pi}$) and consciousness levels in the left orbital middle frontal gyrus and left middle temporal gyrus. Additionally, significant positive correlations were found between consciousness levels and the connectivity variables for pyramidal to excitatory populations ($\alpha_{pe}$) and excitatory to pyramidal populations ($\alpha_{ep}$), primarily in the posterior parietal and occipital regions (Fig. 4b, c). Figure 4d further illustrates positive correlations between

**Table 1 | Significant brain structures and connections related to consciousness levels**

| ROI | p | ROI | p | ROI | p |
|---|---|---|---|---|---|
| | | **Significant brain structures** | | | |
| | | *Significant in-hubs* | | | |
| Frontal_Sup_R | 0.0402 | Frontal_Sup_Medial_R | 0.0021 | Calcarine_L | 0.0253 |
| Cuneus_L | 0.0149 | Cuneus_R | 0.0300 | Lingual_L | 0.0413 |
| Occipital_Sup_L | 0.0097 | Occipital_Sup_R | 0.0440 | Occipital_Mid_R | 0.0067 |
| Occipital_Inf_R | 0.0245 | Parietal_Sup_L | 0.0288 | Parietal_Sup_R | 0.0086 |
| Parietal_Inf_R | 0.0269 | Angular_L | 0.0120 | | |
| | | *Significant out-hubs* | | | |
| Frontal_Inf_Tri_R | 0.0368 | ParaHippocampal_L | 0.0393 | Occipital_Mid_L | 0.0092 |
| Occipital_Mid_R | 0.0058 | Occipital_Inf_L | 0.0172 | Occipital_Inf_R | 0.0251 |
| Parietal_Sup_L | 0.0390 | Angular_L | 0.0430 | | |
| | | **Significant connections** | | | |
| Connection | p | Connection | p | | |
| Occipital_Inf_R → Frontal_Inf_Orb_R | 0.0446 | Parietal_Inf_R → Occipital_Mid_L | 0.0054 | | |
| Calcarine_R → Lingual_R | 0.0469 | Angular_L → Occipital_Mid_L | 0.0232 | | |
| Cuneus_R → Lingual_R | 0.0472 | Angular_R → Occipital_Mid_L | 0.0221 | | |
| Cuneus_L → Occipital_Mid_L | 0.0431 | Occipital_Mid_R → Angular_L | 0.0493 | | |
| Parietal_Sup_L → Occipital_Mid_L | 0.0252 | | | | |

Top panel: significant in-hubs and out-hubs in relation to consciousness levels (i.e., ACPT accuracy) are shown with p values derived from multiple comparisons permutation tests. Bottom panel: significant inter-regional connections correlated with consciousness level are shown with p values derived from multiple comparisons permutation tests. Brain structures are shown in the format of the Automated Anatomical Labeling atlas. Statistics were derived based on $n$ = 15 independent subjects.

external input ($\mu$) and consciousness level in the posterior parietal and occipital regions. Specific brain structures and their corresponding statistics are detailed in Table 4 for each regional variable. These results suggest that the strength of excitatory synaptic connectivity ($\alpha_{ep}$, $\alpha_{pe}$) in the posterior parietal and occipital regions influences consciousness levels in a manner consistent with the expected reduction in excitatory synaptic strength as Xenon concentration increases[49].

As an alternative to the correlation imaging and to provide a comparison against more traditional forms of contrast imaging seen in prior studies, cortical regional variables estimates before and after Xenon equilibration were contrasted. Statistically significant differences were identified using t-tests followed by multiple comparisons permutation tests at $\alpha$ = 0.05 on the t-statistics across all brain regions and subjects. Subjects 5 and 9 were excluded from the analysis as their experiments were terminated before the full 5 minute long Xenon equilibrium period was completed. The significant brain regions are highlighted on the template MRI images. The result emphasizes the prefrontal, including orbitofrontal and dorsal medial frontal areas (Fig. 4e–h, Table 4) as compared to the correlation imaging results.

Specifically, significant decreases were observed in EPSP strength $\alpha_{pe}$ across several regions, including the frontal areas, cingulate gyri, and precuneus. Decreases in $\alpha_{ep}$ were predominantly seen in the posterior parietal and occipital regions, while reductions in $\alpha_{pi}$ were primarily noted in the prefrontal regions. A key difference between contrast and correlation imaging was noted for the external cortical input $\mu$. In the contrast imaging, $\mu$ showed a significant decrease across broader regions, including the frontal cortex, middle and inferior temporal regions, and posterior cortex, whereas correlation imaging highlighted these decreases primarily in the occipital regions. These differences suggest that while some regions are central to both consciousness modulation and Xenon's effects, others play specialized roles in either maintaining consciousness or responding to the drug's pharmacological action.

Additionally, a significant reduction in the membrane potential of pyramidal populations across cerebral cortex was observed, particularly in the posterior parietal and occipital regions, as depicted in Fig. S1.

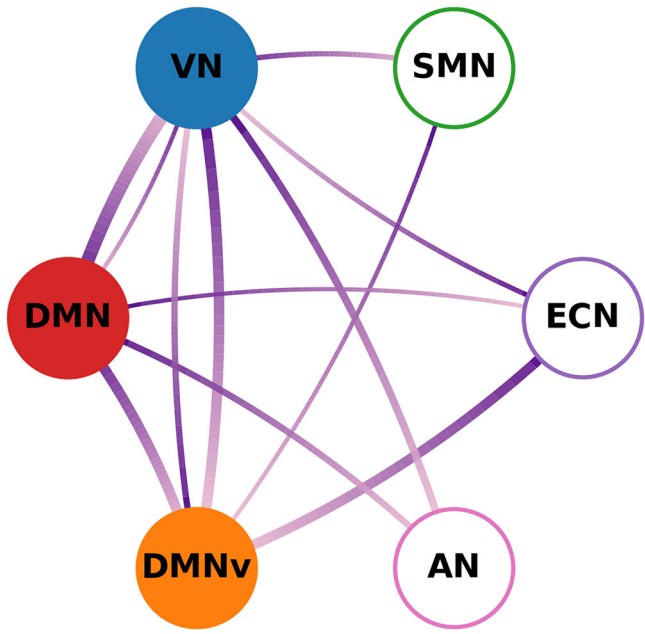

**Fig. 3 | Significant intra- and inter-network connections.** Nodes represent functional networks, with filled nodes indicating those whose intra-network connectivity strength correlates significantly with consciousness level; unfilled nodes denote nonsignificant intra-network correlations. Edges encode directed inter-network connections: each link is drawn as a smooth arc whose width reflects the strength of significance, and whose color runs from deep indigo at the origin (source of influence) to light red at the termination (target of influence).

## Relationship between cortical stability, consciousness levels, and Xenon concentration

As a final analysis cortical stability was correlated with consciousness levels to assess its utility as a global NCC metric and to obtain a deeper

**Table 2 | Intra-network and inter-network connections correlated with consciousness levels**

|  | Network(s) | *p* | Network(s) | *p* | Network(s) | *p* |
|---|---|---|---|---|---|---|
| *Intra-network* | VN | 0.0005 | DMNv | 0.0012 | DMN | 0.0146 |
| *Inter-network* | DMN → VN | 0.0005 | DMN → DMNv | 0.0009 | VN → DMNv | 0.0009 |
|  | ECN → DMNv | 0.0010 | VN → AN | 0.0020 | DMN → AN | 0.0030 |
|  | DMNv → VN | 0.0033 | VN → SMN | 0.0053 | ECN → VN | 0.0064 |
|  | SMN → DMNv | 0.0096 | VN → DMN | 0.0101 | DMN → ECN | 0.0104 |

Networks where the strength of intra-network connections shows significant correlations with consciousness levels (i.e., ACPT accuracy), and connections between networks exhibiting significant correlations with consciousness levels. Arrows indicate the directions of the connections. Permutation tests were applied to find group-level *p* values (see " Methods") and Benjamini-Hochberg procedure was applied to control the FDR at *α* = 0.05. Statistics were derived based on *n* = 15 independent subjects.

**Table 3 | Network connection strength and wSMI correlations with consciousness levels**

| Correlating connection strength to ACPT accuracy | | Correlating wSMI to ACPT accuracy | |
|---|---|---|---|
| Network(s) | *p* | Network(s) | *p* |
| Posterior parietal | 0.0475 | Posterior parietal | 0.0212 |
| Posterior parietal → Prefrontal | 0.0495 | Posterior parietal → Prefrontal | 0.0397 |
|  |  | Whole brain | 0.0191 |

Arrows indicate the connections between two networks. Permutation tests were applied to find group-level *p* values (see " Methods"). Statistics were derived based on *n* = 15 independent subjects.

understanding within the context of dynamical systems theory. In this study, cortical stability corresponds to linear stability of the whole-cortex model. To quantify cortical stability, the eigenvalues of the Jacobian matrix derived from the whole-cortex model were analyzed. The Jacobian matrix was computed using model parameters estimated from Xenon-induced anesthesia data. Specifically, parameters were estimated within successive one-second time windows, capturing the temporal dynamics of the cortical system under varying levels of consciousness. These time-specific parameter estimates were then used to construct a Jacobian matrix for each time window, allowing us to examine the temporal evolution of cortical stability throughout the recording. The stability of the system was assessed by analyzing the eigenvalues of the Jacobian matrix in each time window. Eigenvalues with negative real parts indicate a stable cortical state, while eigenvalues with positive real parts signify instability. The number of unstable eigenmodes, represented by eigenvalues with positive real parts, was used as a quantitative marker of cortical stability in this study. Each unstable eigenmode corresponds to a dimension in the system's state space where small perturbations grow over time, driving the system toward instability. The count of these eigenmodes reflects the degree to which the system deviates from equilibrium.

Figure 5a illustrates the changes in cortical stability over time for individual subjects undergoing Xenon-induced anesthesia. Firstly, consistent cortical instability was observed across all subjects, indicating that the cerebral cortex remained in an unstable state throughout the duration of the Xenon-induced anesthesia. Secondly, the temporal patterns depicted in the figure reveal correlations between the number of unstable eigenmodes and consciousness levels. Specifically, a decrease in the number aligns with a reduction in consciousness, reflecting the stabilization of cortical dynamics. In contrast, a rising number signifies recovery phases, reflecting a shift towards destabilization of cortical activity. Notable subject-specific variations are apparent. Subject 14 displays a persistently constant number of unstable eigenmodes throughout much of the experiment. In contrast, subject 15 shows a rapid decline in instability associated with the increase of Xenon concentration levels rather than consciousness levels. The statistical analysis in Table 5 supports and extends these observations. Significant positive correlations

between cortical stability and consciousness levels were observed for 13 of 15 subjects ($p < 0.05$), indicating that greater cortical instability corresponds to reduced consciousness. However, individual variability is evident, as some subjects, such as subject 6, exhibit negligible correlations, suggesting that the relationship between cortical stability and consciousness levels is less pronounced or inconsistent in these cases. Moreover, the trajectories of unstable eigenmodes reveal transient fluctuations during both induction and recovery phases, suggesting dynamic transitions rather than abrupt state changes. These fluctuations represent intermediate cortical states as the brain transitions between less unstable and more unstable regimes.

Another important observation is that while cortical stability exhibits a significant correlation with Xenon concentration levels, its relationship with consciousness levels is notably stronger. This distinction is evident in both Fig. 5a and the correlation analyses in Table 5. For most subjects, the absolute correlation coefficients between cortical stability and consciousness levels are higher than those with Xenon concentration levels. This suggests that cortical stability, as reflected by the number of unstable eigenmodes, is more closely aligned with changes in consciousness levels.

A notable dissociation between the level of consciousness and Xenon concentration levels is evident in both the figure and the statistical results. While Xenon concentration levels increase steadily during the induction phase, the corresponding decrease in consciousness levels often exhibits a time delay. This lag suggests that the pharmacological effects of Xenon on cortical activity require a critical threshold or cumulative exposure before manifesting as observable changes in consciousness levels. For example, in several subjects, such as subject 18, the level of consciousness remains relatively stable for a period despite rising Xenon concentration, before eventually declining sharply.

Figure 5b summarizes the within-subject Pearson correlation coefficients for the three variable pairings across all 15 subjects. The boxplot on the left shows that correlations between cortical stability and consciousness level are generally positive (median ≈ 0.4, IQR ≈ 0.5), indicating a strong coupling of instability with loss of consciousness. By contrast, the middle boxplot (cortical stability vs. Xenon concentration) is centered slightly below zero, and the right boxplot (consciousness level vs. Xenon concentration) exhibits consistently negative correlations (median ≈ –0.2, IQR ≈ 0.4). Together, these results confirm that cortical stability tracks changes in consciousness more closely than it does Xenon concentration, and that high Xenon concentration level is accompanied by loss of consciousness.

Additionally, substantial inter-subject variability is also apparent in the duration and degree of loss of consciousness. Some subjects, such as subjects 2 and 11, experienced prolonged periods of unconsciousness, whereas other subjects, such as subject 6, demonstrated relatively brief or incomplete loss of consciousness. This variability highlights individual differences in susceptibility to Xenon-induced anesthesia, potentially reflecting underlying neurophysiological or pharmacokinetic differences.

## Discussion
This study aimed to identify critical neurobiological correlates of consciousness and establish a framework to explain the neural mechanisms

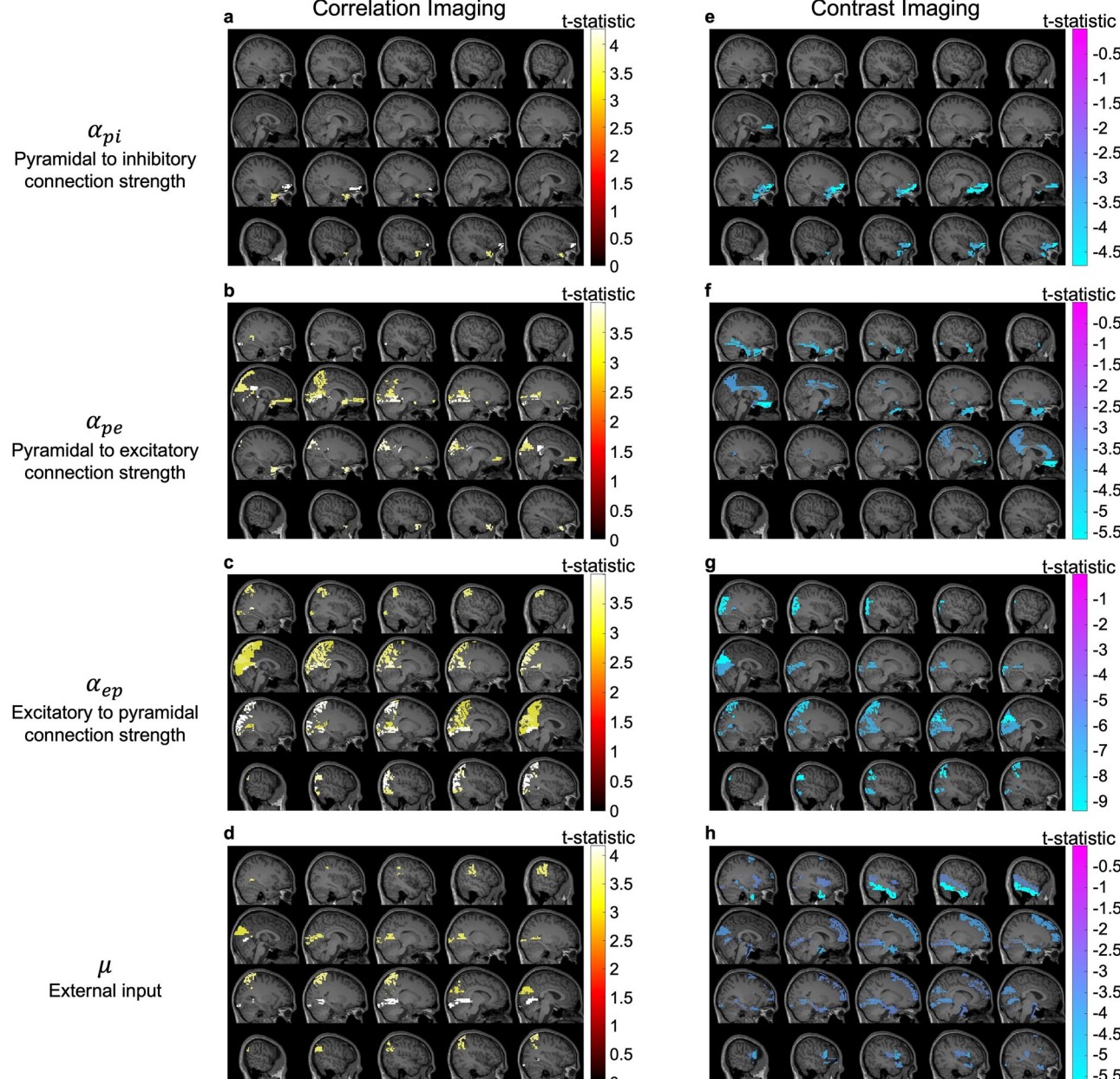

**Fig. 4 | Brain structures showing significant correlation between consciousness level (i.e. ACPT accuracy) and regional neurophysiological variable estimates.** **a–d** Correlation imaging reveals cortical regions exhibiting notable correlations between consciousness levels and specific regional variable estimates. **e–h** Contrast imaging shows cortical regions demonstrating significant differences in specific regional variable estimates between the awake ACPT period and the equilibrated Xenon concentration period.

which are correlated with, and likely also regulate, the level of global consciousness. To achieve this a whole-cerebral-cortex model employing interconnected NMMs was proposed, with each NMM representing a specific region of interest in the cerebral cortex. We are aware of neural mass models that split inhibition into fast parvalbumin (PV) and slow somatostatin (SST) pools that would allow a richer exploration of frequency-specific mechanisms[50]. However, many frameworks that probe wakefulness, anesthesia and disorders of consciousness keep a single composite inhibitory population[51–54]. Adopting this simple arrangement keeps the parameter space compact and ensures that the parameters we estimate from non-invasive MEG remain identifiable, while still capturing the large-scale stability shifts that accompany conscious state changes. Moreover, Xenon is a putative NMDA receptor antagonist, as such its predominant mode of action is on excitatory rather than inhibitory synapses. This makes it less

necessary to include additional inhibitory populations. In future work the model could be explicitly extended to include PV and SST populations and test whether this added detail improves performance across different brain conditions and other GABA receptor agonist-based anesthetics. Crucially, the forward generative nature of the model is itself instrumental. It yields latent neurophysiological variables-region-specific synaptic gains and the eigenvalues of the system Jacobian that cannot be obtained with purely descriptive connectivity measures. Techniques such as Granger causality test linear predictability between observed time-series but neither embody the nonlinear circuit dynamics nor avoid the sensitivity to filtering and volume conduction artifacts that cloud mechanistic interpretation[55]. By fitting our model to reproduce the empirical MEG spectra, we can recover those hidden neurophysiological parameters and state variables allowing for relating neurophysiological variables to the change of consciousness levels.

**Table 4 | Brain regions with significant t-statistics in correlation and contrast imaging**

| | $a_{pi}$ | t-statistic | $a_{pe}$ | t-statistic | $a_{ep}$ | t-statistic | $\mu$ | t-statistic |
|---|---|---|---|---|---|---|---|---|
| *Correlation* | Frontal_Mid_Orb_L | 4.2717 | Cingulum_Post_L | 4.0126 | Occipital_Mid_L | 4.0056 | Lingual_L | 4.1726 |
| *Imaging* | Temporal_Pole_Mid_L | 3.7224 | Lingual_R | 3.8917 | Parietal_Sup_L | 3.9389 | Parietal_Sup_L | 3.7324 |
| | | | Occipital_Sup_L | 3.8459 | Lingual_L | 3.9112 | SupraMarginal_R | 3.6124 |
| | | | Temporal_Pole_Mid_L | 3.7168 | Calcarine_R | 3.8012 | Calcarine_R | 3.5762 |
| | | | Olfactory_R | 3.5871 | Angular_L | 3.7478 | Angular_L | 3.5735 |
| | | | Cuneus_L | 3.5177 | Occipital_Sup_R | 3.7282 | Cuneus_L | 3.3911 |
| | | | Calcarine_R | 3.4504 | Cingulum_Post_R | 3.6354 | | |
| | | | Frontal_Med_Orb_R | 3.4348 | Occipital_Inf_L | 3.5362 | | |
| | | | Frontal_Med_Orb_L | 3.4132 | Precuneus_R | 3.5353 | | |
| | | | Precuneus_R | 3.4036 | Cuneus_R | 3.5109 | | |
| | | | | | Cuneus_L | 3.4961 | | |
| | | | | | Parietal_Inf_R | 3.4857 | | |
| | | | | | Parietal_Sup_R | 3.4843 | | |
| | | | | | Occipital_Inf_R | 3.3933 | | |
| | | | | | Precuneus_L | 3.2482 | | |
| | | | | | Paracentral_Lobule_R | 3.2410 | | |
| | | | | | Calcarine_L | 3.2163 | | |
| *Contrast* | Frontal_Sup_Orb_L | −4.7673 | Rectus_L | −5.6424 | Angular_L | −9.3609 | Temporal_Inf_R | −5.6335 |
| *Imaging* | Frontal_Mid_Orb_L | −4.7284 | Temporal_Pole_Mid_R | −4.8823 | Cuneus_L | −9.2590 | Frontal_Inf_Oper_L | −4.3361 |
| | Frontal_Med_Orb_L | −4.1911 | Fusiform_R | −4.6501 | Occipital_Mid_R | −9.0837 | ParaHippocampal_R | −4.2258 |
| | Temporal_Pole_Mid_L | −4.0599 | Cingulum_Ant_L | −4.1818 | Occipital_Inf_R | −8.7592 | Frontal_Sup_R | −3.9891 |
| | Frontal_Inf_Orb_L | −3.7773 | Cingulum_Post_L | −4.1181 | Parietal_Sup_L | −8.4166 | Temporal_Pole_Sup_L | −3.8030 |
| | | | Cingulum_Mid_R | −4.1059 | Occipital_Inf_L | −7.8248 | Lingual_L | −3.8016 |
| | | | Olfactory_R | −3.9628 | Calcarine_R | −7.3398 | Cuneus_L | −3.7809 |
| | | | Precuneus_L | −3.9040 | Lingual_L | −7.1325 | Frontal_Sup_Medial_R | −3.5087 |
| | | | | | Calcarine_L | −7.0919 | Rolandic_Oper_L | −3.4915 |
| | | | | | Occipital_Sup_L | −7.0552 | Heschl_L | −3.4770 |
| | | | | | | | Temporal_Mid_R | −3.4651 |
| | | | | | | | Frontal_Sup_L | −3.4626 |
| | | | | | | | ParaHippocampal_L | −3.4271 |
| | | | | | | | Lingual_R | −3.4081 |
| | | | | | | | Insula_R | −3.3862 |

Statistics were derived based on *n* = 15 independent subjects.

The model is therefore not a mere convenience but the analytical lens through which the stability shifts associated with conscious-state transitions become visible.

Specifically, the model parameters consist of regional neurophysiological variables and inter-regional connectivity, which were estimated from source-level MEG time series of healthy human subjects exposed to varying concentration levels of Xenon to induce loss of consciousness/responsiveness and recover from it. The level of consciousness was characterised by a subject's responsiveness level measured by ACPT accuracy (Fig. 1c)[39]. To identify the critical NCCs, the relationship between consciousness/responsiveness levels and the time-evolving parameter estimates of the whole-cortex model was examined. The key findings are as follows.

First, correlation between consciousness levels and centrality measures of brain regions in the posterior parietal and occipital regions was identified (Fig. 2a and b). Besides, there are hitherto few studies underscoring posterior parietal-to-occipital connectivity in the reduction of consciousness[19,56]. However, in Fig. 2c, we demonstrated that the strengths of several connections from brain structures in the parietal lobe to those in the occipital lobe are significantly correlated with the level of consciousness. Second, strong correlations were found between consciousness levels and the

strength of connections from posterior parietal to prefrontal regions (Table 2). This suggests that the flow of information or activity from the posterior parietal areas (involved in processes such as spatial attention and awareness of the body's position in space) to the prefrontal cortex (responsible for complex cognitive behavior, decision-making, and moderating social behavior) is correlated with the level of consciousness. The posterior parietal region provides the prefrontal cortex with sensory and spatial information necessary for conscious awareness and higher-order cognitive processes. This pattern of unidirectional connectivity suggests that the posterior parietal region's role in consciousness is not as a receiver of processed information from the prefrontal cortex but rather as an active contributor to the conscious experience, possibly by integrating sensory information and relaying it forward.

Third, consciousness levels were significantly correlated with intra-posterior parietal connection strength, but not with intra-prefrontal connections (Table 2). This underscores the significance of internal connectivity within the posterior parietal area for sustaining global conscious states. The absence of a significant correlation within the prefrontal cortex suggests that while it is important for higher-order cognitive functions and decision-making, the strength of its internal connections is not influential on the level

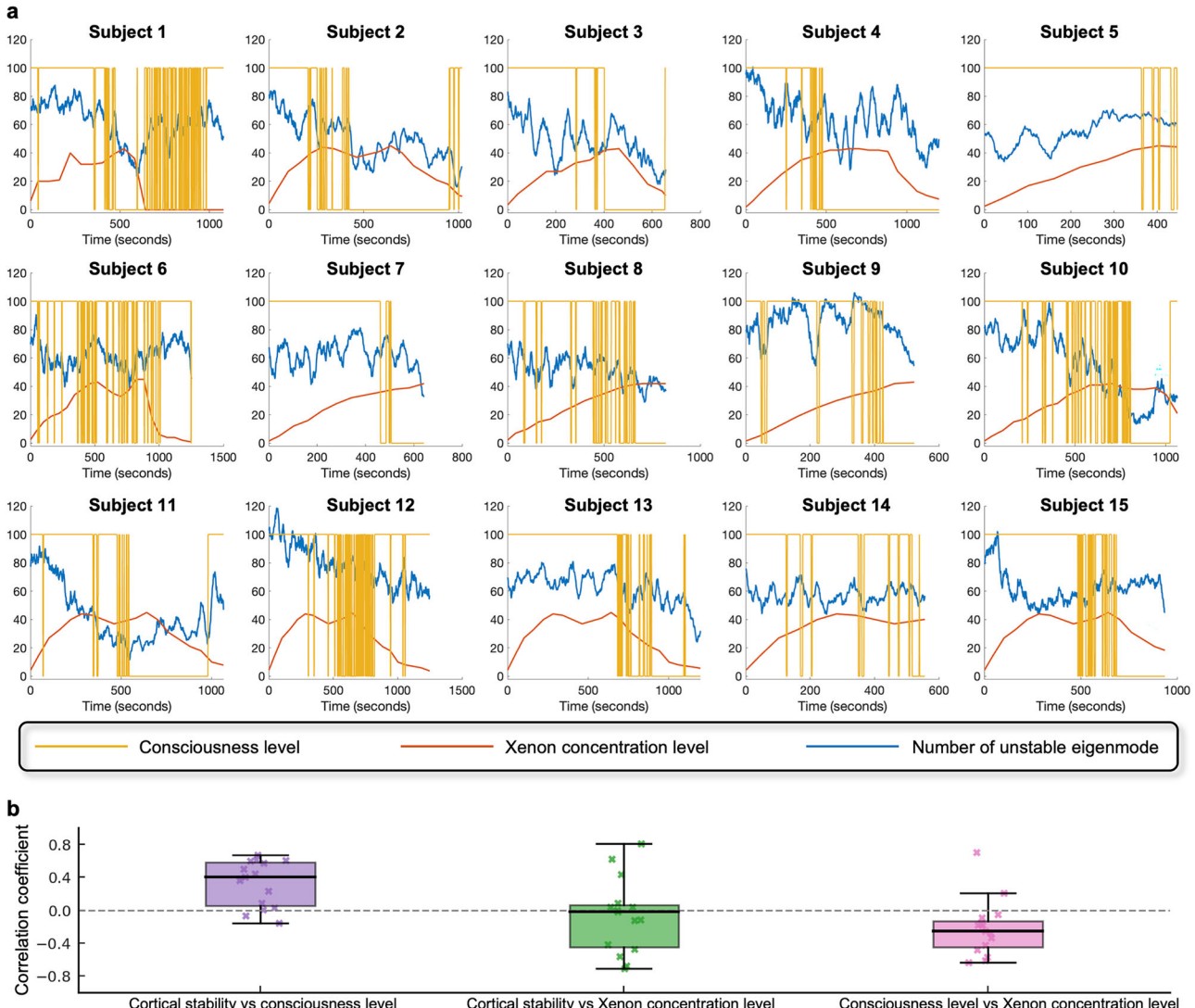

**Fig. 5 | Time courses of cortical stability, consciousness level, and Xenon concentration and their within subject correlations. a** Cortical stability is represented by the number of unstable eigenvalues (blue line). The yellow lines indicate the level of consciousness/responsiveness (i.e., ACPT accuracy), with 100 denoting conscious state and 0 denoting unconscious state. Red lines represent the concentration level of Xenon anesthesia administered over time. **b** Boxplots summarizing Pearson correlation coefficients across subjects for three pairings: cortical stability vs consciousness level; cortical stability vs Xenon concentration level; and consciousness level vs Xenon concentration level. Boxes span the interquartile range (median line), whiskers denote the full range, and individual subject values are overlaid as jittered points.

of consciousness. This is consistent with previous studies on brain lesion patients who sustained bilateral prefrontal cortex. While they are impaired in social and emotional domains, overall consciousness, especially in the sense of sensory and qualitative aspects, was remarkably preserved[57–61].

Fourth, it was shown in Fig. 4 that regional variables in the posterior parietal and occipital regions were significantly positively correlated with consciousness levels. Low-level consciousness was associated with reduced regional excitatory synaptic connection strength ($\alpha_{pe}$, $\alpha_{ep}$), diminished afferent input ($\mu$) and lowered membrane potentials of pyramidal neuronal populations in those regions. This observation not only aligns with existing literature demonstrating that Xenon attenuates excitatory synaptic transmission[49,62] but also highlights how this effect varies in relation to consciousness levels across the cerebral cortex. Collectively, while the integrated information theory suggests that the posterior brain regions are critical for consciousness due to the grid-like structures in the visual cortex[63], which are highly efficient at generating large amounts of integrated information, our study independently supports this conclusion by analyzing the neurophysiological processes associated with consciousness.

Fifth, critical regional variable estimates were mapped onto functional networks in Table S3. Results showed that regional excitatory synaptic connection parameters in VN, DMN, and executive control network (ECN) decreased during reductions in consciousness, concurring with attenuation of the afferent input in all considered functional networks. While prior research has underscored the importance of connectivity alterations within functional networks across diverse levels of consciousness[24,26,64], this study advances understanding by pinpointing the precise neurophysiological variables within these networks that underpin varying levels of consciousness.

In the section titled Relationship between cortical stability, consciousness levels, and Xenon concentration, we quantified cortical stability by counting, at every 1-s window, how many eigenvalues of the time-resolved whole-cortex Jacobian had positive real part. Evaluating cortical stability via the number of positive eigenvalues is an increasingly common approach in the linear stability analysis in brain modeling[65,66]. Because every positive eigenvalue marks an independent phase space direction in which infinitesimal perturbations grow, the number or proportion of such modes

**Table 5 | Within-subject correlations between cortical stability, consciousness, and Xenon concentration levels**

| | Cortical stability vs consciousness level | | Cortical stability vs Xenon concentration level | | Consciousness level vs Xenon concentration level | |
|---|---|---|---|---|---|---|
| | Correlation coefficient | p | Correlation coefficient | p | Correlation coefficient | p |
| Subject 1 | 0.5930 | <0.0001 | 0.0397 | 0.1922 | −0.0510 | 0.0940 |
| Subject 2 | 0.4411 | <0.0001 | 0.0380 | 0.2256 | −0.4221 | <0.0001 |
| Subject 3 | 0.3607 | <0.0001 | −0.0046 | 0.9066 | −0.1828 | <0.0001 |
| Subject 4 | 0.4941 | <0.0001 | −0.1255 | <0.0001 | −0.3344 | <0.0001 |
| Subject 5 | −0.1578 | <0.0001 | 0.8057 | <0.0001 | −0.2521 | <0.0001 |
| Subject 6 | 0.0252 | 0.3726 | −0.4206 | <0.0001 | −0.1839 | <0.0001 |
| Subject 7 | 0.2298 | <0.0001 | 0.0837 | 0.0341 | −0.6099 | <0.0001 |
| Subject 8 | 0.5684 | <0.0001 | −0.5621 | <0.0001 | −0.6321 | <0.0001 |
| Subject 9 | 0.4014 | <0.0001 | −0.0216 | 0.6221 | −0.5739 | <0.0001 |
| Subject 10 | 0.6081 | <0.0001 | −0.6730 | <0.0001 | −0.4792 | <0.0001 |
| Subject 11 | 0.6058 | <0.0001 | −0.4715 | <0.0001 | −0.0944 | 0.0020 |
| Subject 12 | 0.0818 | 0.0038 | 0.4340 | <0.0001 | −0.2785 | <0.0001 |
| Subject 13 | 0.6672 | <0.0001 | 0.6236 | <0.0001 | 0.6973 | <0.0001 |
| Subject 14 | 0.0021 | 0.9615 | −0.1149 | 0.0067 | −0.1644 | <0.0001 |
| Subject 15 | -0.0696 | 0.0333 | −0.7059 | <0.0001 | 0.2055 | <0.0001 |

Correlation coefficients and corresponding $p$ values for the possible pairings of cortical stability, consciousness levels (i.e., ACPT accuracy), and Xenon concentration levels for each individual subject are shown. Statistics were derived based on $n = 15$ independent subjects.

provides a scale-free, analytically tractable index of how many " routes to instability" the cortex affords, derived directly from fitted biophysical parameters, without lengthy simulations, and thus offers an interpretable and useful gauge for comparing stability across wakefulness, anesthesia and pathology. In canonical dynamical systems terms, a change in the sign of the real part of a complex-conjugate pair of eigenvalues marks a (super- or sub-critical) Hopf bifurcation, beyond which a small amplitude limit cycle is born. A large body of whole-brain modeling work has argued that resting brain dynamics operate in the subcritical regime, i.e., just below this Hopf point, because this maximizes metastability, information transfer capability and susceptibility to perturbations[67–69]. Our results are consistent with that view. When participants were behaviorally responsive the cortex showed a moderate number of unstable eigenmodes, indicating that many regions sit close to-yet just on the unstable side of-the Hopf boundary. As consciousness waned under Xenon, the number of unstable modes fell, implying that the effective control parameters (e.g., local excitatory gain) moved the system deeper into the linearly stable half-plane and therefore farther from the Hopf surface. Conversely, during recovery the eigenvalues drifted back towards the imaginary axis and the count of positive-real modes rose, representing a return toward the optimal, near-critical working point. In other words, the " bifurcation landscape" itself appears to shift over time with arousal level, rather than remaining fixed while the brain merely " chooses" different attractors within it. A bifurcation analysis with our model is intractable because there are too many free parameters (i.e., 78 local models × 3 excitatory connection strengths + 78 × 78 inter-regional connection strengths), but the temporally resolved eigenvalue approach used here still captures when the system crosses local stability boundaries. Future work could project the high-dimensional parameter trajectory onto one or two physiologically interpretable summary variables, such as the mean excitatory synaptic gain or a global coupling constant, and then perform classic bifurcation analysis to visualize the Hopf surface[70]. Nevertheless, the present results can link the stability metric to the Hopf framework and show that loss (and recovery) of consciousness corresponds to systematic excursions in this bifurcation landscape.

In Fig. 5, it was found that cerebral cortex never stabilized during reductions of consciousness, extending the idea of instability of resting brain networks[71] while supporting the observation obtained from monkey subjects[65]. Besides, previous studies have established that loss of consciousness is often associated with stabilized neural dynamics, as measured through decreases in brain complexity and increases in low-frequency power[72,73]. Our results corroborate these findings by demonstrating that reductions in the number of unstable eigenmodes are associated with lower levels of consciousness in most subjects. This reflects a stabilization of cortical activity during unconscious states, suggesting that the brain transitions toward a dynamically constrained regime under Xenon anesthesia. However, the transient fluctuations observed during induction and recovery phases offer a more nuanced view than the abrupt state changes reported in some prior literature[74,75], emphasizing the dynamic and gradual nature of these transitions. Another notable contribution of this study is the dissociation observed between Xenon concentration levels and consciousness levels, highlighting individual pharmacodynamic differences. While prior research has indicated that anesthetic effects on consciousness often follow nonlinear dose-response relationships[76], our findings extend this understanding by demonstrating a temporal disassociation between Xenon concentration increases and reductions in consciousness. Moreover, the substantial inter-subject variability aligns with existing evidence of heterogeneity in anesthetic responses[77] but further underscores the importance of individual neurophysiological and pharmacokinetic factors in determining the trajectory of consciousness loss and recovery.

Several frameworks already linked whole-brain models to neuroimaging data. The Virtual Brain (VB) fits neural mass equations onto an individual's diffusion-MRI connectome to reproduce empirical EEG/MEG signatures. In its canonical form the anatomical weights are fixed, and only a handful of global control variables are tuned until simulated spectra and static functional-connectivity match the data[78]. Dynamic mean-field (DMF) formulations compress each region to an excitation inhibition pair whose collective firing rate is then coupled through the empirical connectome, and a single global coupling constant is iteratively adjusted until the model reproduces time-averaged fMRI correlations[79]. Further simplification leads to Hopf/Stuart-Landau oscillators poised near a super-critical bifurcation, which explain resting-state " turbulence" and state-dependent spectral shifts with only a regional bifurcation parameter and a global coupling gain[69]. A variant is the Kuramoto phase model where each area contributes a single phase variable, time delays approximate axonal conduction, and large-scale synchrony emerges from sinusoidal coupling, successfully recapitulating slow

https://doi.org/10.1038/s42003-025-08782-6                                                                                            **Article**

BOLD co-fluctuations at minimal computational cost[11]. These frameworks illuminate universal dynamical principles but ignore distinct synapse classes, assume the structural connectome is immutable, and usually infer no more than one or two global parameters. Our framework inherits VB's commitment to anatomical realism while addressing the above limitations. Each cortical node is a biologically plausible Jansen-Rit neural mass that retains separate pyramidal, excitatory and inhibitory populations. Directly from the empirical MEG timeseries, we estimate time-resolved local synaptic gains, afferent drives and inter-regional coupling strengths. Consequently, the local and inter-regional connection strengths are allowed to drift with conscious states. By recovering multiple neurophysiological variables per region, the framework allows correlating neurophysiological variables to the change of brain conditions in the time-domain and identifies specific brain regions that are responsible for the change. These kinds of inference analyses have not yet been studied with the VB and oscillator-based model frameworks.

The biological inferences reported here are not an artifact of the particular " starter" parametrisation we adopted but instead withstand several layers of quantitative and conceptual analyses. First, the analytic Kalman filter that anchors the pipeline tracks MEG time series with subject-wise RMSEs that sit within 10% of each signal's dynamic range (Fig. S2), while the subsequent multivariate regression that estimates the inter-regional weights explains 89% of the variance of the empirical inputs in every 1-s window. Second, the same Kalman-filter and NMM architecture reliably reproduces resting alpha activity in healthy cortex[37] and captures the rapid transition from interictal to ictal phase in focal epilepsy[42]. The underlying Jansen-Rit family of models has long been used to generate EEG dynamics observed in vivo[80], confirming the model is sufficiently expressive to capture diverse cortical dynamics, and provides a reasonable trade-off between biological realism and model complexity to enable efficient and less degenerate model inference. Third, all statistical claims were supported by permutation tests with family-wise error control, which reproduced the same qualitative significance pattern when the signs of correlation coefficients were randomly flipped. Nonetheless, modeling limitations should be acknowledged regarding to the robustness of our findings. Every cortical area is currently represented by the same canonical three-population Jansen-Rit NMM. Richer region-specific NMMs that better capture local resting spectra (e.g., with distinct PV and SST interneuron pools) can be swapped in without altering the inference method. Thalamic drivers and finer synaptic-time-constant heterogeneity are omitted. Because the framework is fully generative and modular, these extensions can be incorporated in future work to confirm that the principal conclusions are insensitive to such biophysical detail while giving an even tighter account of regional dynamics.

In this study, Xenon was employed to modulate the subject's global consciousness levels. Future research should consider utilizing a range of anesthetics that target different molecular pathways, offering additional universal insights into network connectivity, local dynamics, and cortical stability across various consciousness levels. Moreover, although we treat responsiveness and global consciousness interchangeably in this paper, future work should be directed at using this framework to better understand NCCs by disentangling ideas around responsiveness, connectedness, and consciousness[81]. We also recognize that our analysis focused on how neurophysiological variables relate to the level of consciousness and did not explicitly test their relationship with Xenon concentration levels. We therefore acknowledge this as a study limitation and plan to explore dose dependent effects in future work by acquiring higher resolution Xenon measurements and assessing how parameter estimates and state trajectories scale with inhaled concentration. This framework can be applied to various brain conditions (e.g., sleep[82], brain injury, deep anesthesia) and extended to animal models[83,84], offering insights into conserved neurophysiological mechanisms and dynamic cortical behaviors to advance understanding of NCCs and brain function.

## Methods
### Dataset
The present study utilized data collected from previous research that investigated changes in EEG/MEG signals from 15 subjects during Xenon-anesthetic-induced reductions of consciousness. For details, see[39]. Some key points of the data are summarized below.

Male participants, aged between 20 and 40 years, who were right-handed and had a body mass index (BMI) within the range of 18–30, were recruited for the study. Female volunteers were excluded to avoid potential effects of pregnancy on EEG readings during anesthesia. Individuals with a history of neurological disorders, epilepsy, mental health conditions, cardiovascular or respiratory diseases, motion sickness, asthma, claustrophobia, obstructive sleep apnea, or recent use of prescription, recreational, or psychoactive drugs were not eligible to participate.

During the recording session, participants were asked to keep their eyes closed and remain still. Neural signals were recorded using a 306-channel MEG system (Elekta Neuromag TRIUX, Elekta Oy, Finland) at a sampling rate of 1000 Hz.

MEG recordings were obtained during Xenon-induced anesthesia at a concentration of 1.3 MAC-awake (42% $Xe/O_2$). Prior to the induction of anesthesia, participants and the attending nurse or clinical observer were informed about the start of gas administration. MEG data collection was initiated simultaneously with the anesthesiologist beginning gas delivery, marking the onset of the experiment. Throughout the session, responsiveness was monitored using the auditory continuous performance task (ACPT). Participants wore MEG-compatible headphones that delivered binaural auditory tones at fixed stereo amplitude with frequencies of either 1 kHz or 3 kHz. The tones were presented at random intervals ranging between 2 and 4 seconds. Participants were directed to respond promptly by pressing buttons on two handheld devices, using the left button to indicate low-frequency tones and the right button for high-frequency tones. The accuracy of responses in the ACPT was used as a measure of responsiveness, where a correct response was scored as 100, and a failure to respond or an incorrect response (rarely occurring) was scored as 0.

A concentration of 42% $Xe/O_2$ was chosen to induce a loss of responsiveness in most participants, as supported by previous research[85]. Higher concentrations were avoided since the study was conducted outside a hospital setting, necessitating strict adherence to safety protocols as outlined in the ethics approval. The Xenon concentration was manually adjusted to reach the target equilibrium and maintained at that level until the anesthesiologist or clinical observer deemed it unsafe to proceed. At this point, the gas administration was terminated, and a washout phase with 100% $O_2$ was initiated.

To ensure the reliability of the MEG data, artifact removal and signal preprocessing were performed. This process included notch filtering to eliminate power line noise and additional steps to address artifacts, such as those caused by eye movements. These measures were essential to improve the quality of the recordings and reduce interference from non-neuronal sources, facilitating accurate analysis. For source reconstruction, an atlas-guided approach employing the linearly constrained minimum variance (LCMV) beamformer method was utilized[37,86]. To mitigate the depth bias typically associated with this technique, the beamformer weights were adjusted through normalization[87]. Given that MEG predominantly captures signals from superficial brain regions, the analysis was confined to the gray matter of the cerebral cortex. Specifically, it examined 78 voxels, each representing the source voxel with the highest power within one of 78 brain regions defined by the AAL atlas in the cerebral cortex[41].

### Jansen-Rit neural mass model
The canonical Jansen-Rit neural mass model (NMM)[38] was used in this study to represent the 78 cortical structures of the brain at a macroscopic scale. This model has been extensively utilized in prior research[37,54,88–90] and is particularly suitable for simulating source-level MEG signals. Figure 1a depicts the schematic of the NMM, which consists of pyramidal neurons, excitatory interneurons, and inhibitory interneurons.

The pyramidal population receives input $\mu$ and sends signals to both the excitatory and inhibitory populations. These populations, in turn, provide feedback to the pyramidal neurons, with excitatory neurons delivering positive feedback and inhibitory neurons contributing negative feedback. The model represents the dynamics of each population by their mean membrane potential $v_n$, which evolves over time due to cumulative post-synaptic membrane potentials $v_{mn}$, mediated by synaptic connections. Each synapse is characterized by a connection strength parameter $\alpha_{mn}$. In this context, $m$ denotes the pre-synaptic neuronal population and $n$ refers to the post-synaptic neuronal population.

The post-synaptic membrane potentials are defined as

$$v_{mn}(t) = \alpha_{mn} \int_0^t h_{mn}(t - t')g(v_m(t'))\, dt', \tag{1}$$

where the post-synaptic response kernel $h_{ba}(t)$ is defined as

$$h_{mn}(t) = \eta(t)\frac{t}{\tau_{mn}}\exp\left(-\frac{t}{\tau_{mn}}\right), \tag{2}$$

and $\eta(\cdot)$ is Heaviside step function. The integral can be expressed as coupled first-order differential equations

$$
\begin{aligned}
\frac{dv_{mn}}{dt} &= z_{mn},\\
\frac{dz_{mn}}{dt} &= \alpha_{mn}g(v_m) - \frac{2}{\tau_{mn}}z_{mn} - \frac{1}{\tau_{mn}^2}v_{mn},
\end{aligned}
\tag{3}
$$

where $\tau_{mn}$ is the synaptic time constant, and $\alpha_{mn}$ is the connection strength. The nonlinear firing rate function $g(v_m)$ has various forms and this study used a sigmoid function as suggested by some previous studies[37,54,89]

$$g(v_m) = \frac{1}{2}\left(\mathrm{erf}\left(\frac{v_m - v_0}{\sqrt{2}\varsigma}\right) + 1\right), \tag{4}$$

where $v_0$ and $\varsigma$ represent the mean and variance of firing thresholds, respectively[88,91,92]. The pyramidal membrane potential is modeled as all contributing signals

$$v_p(t) = \mu(t) + v_{ip}(t) + v_{ep}(t), \tag{5}$$

which is corresponding to the recorded source-level MEG signals.

The model parameter vector is

$$\theta = [\mu, \alpha_{ip}, \alpha_{pi}, \alpha_{pe}, \alpha_{ep}]^\top, \tag{6}$$

while the state vector captures the fast-changing dynamics

$$\mathbf{x} = [v_{ip}, z_{ip}, v_{pi}, z_{pi}, v_{pe}, z_{pe}, v_{ep}, z_{ep}]^\top. \tag{7}$$

They are concatenated into an extended state vector

$$\xi = [\mathbf{x}^\top, \theta^\top]^\top. \tag{8}$$

The continuous-time system is defined by the following equation

$$\dot{\xi} = \mathbf{A}\xi + \mathbf{B}\xi \circ g(\mathbf{C}\xi), \tag{9}$$

where $\circ$ represents element-wise multiplication. To estimate the parameters, the model is discretized and incorporates a noise term $\mathbf{w}_t$ accounting for process noise and modeling error

$$\xi_{t+1} = \mathbf{A}_\delta\xi_t + \mathbf{B}_\delta\xi_t \circ g(\mathbf{C}_\delta\xi_t) + \mathbf{w}_t. \tag{10}$$

Here, $\mathbf{A}$, $\mathbf{B}$, and $\mathbf{C}$ are defined in Supplementary Note.

The measurement model links the observed MEG time series to the averaged membrane potential of pyramidal population

$$y_t = \mathbf{H}\xi_t + v_t. \tag{11}$$

Here, $\mathbf{H}$ is the measurement matrix, and $v_t$ is Gaussian white noise representing measurement noise. The matrix $\mathbf{H}$ computes the pyramidal membrane potential as specified in Eq. (5).

## Whole-cortex model

The coupling between cortical regions is modeled by connecting the output firing activity of the pyramidal population in region $a$ to the input of the pyramidal population in region $b$ via a post-synaptic response kernel. Pyramidal populations integrate post-synaptic membrane potentials induced by firing rates of all connected cortical regions. The parameter $w_{ab}$ represents the connectivity strength between region $a$ and region $b$, capturing inter-regional interactions rather than local synaptic dynamics. This hierarchical organization allows for a biologically plausible representation of regional interactions.

For a network comprising $N$ cortical regions (where $N = 78$ in this study because there are 78 brain structures considered in the whole-cortex model), the neural mass model (NMM) for a single region $b$ is defined by its state vector

$$\mathbf{x}^b = [\mu^b, u^b, v_{ip}^b, z_{ip}^b, v_{pi}^b, z_{pi}^b, v_{pe}^b, z_{pe}^b, v_{ep}^b, z_{ep}^b]^\top, \tag{12}$$

where $v$ and $z$ denote the mean membrane potential of a particular type of neuronal population and its time derivative, respectively. The input $\mu^b$ captures the summation of post-synaptic membrane potentials from all connected regions (including self-connection), and $u^b$ represents its rate of change.

The temporal evolution of neural mass model states in region $b$ is defined as

$$
\begin{aligned}
\frac{d\mu^b}{dt} &= u^b, & \frac{du^b}{dt} &= \sum_{b=1}^N w_{ab}g(\mu^a + v_{ep}^a + v_{ip}^a) - \frac{2}{\tau_d}u^b - \frac{1}{\tau_d^2}\mu^b,\\
\frac{dv_{ip}^b}{dt} &= z_{ip}^b, & \frac{dz_{ip}^b}{dt} &= \alpha_{ip}^b g(v_{pi}^b) - \frac{2}{\tau_{ip}}z_{ip}^b - \frac{1}{\tau_{ip}^2}v_{ip}^b,\\
\frac{dv_{pi}^b}{dt} &= z_{pi}^b, & \frac{dz_{pi}^b}{dt} &= \alpha_{pi}^b g(v_{ip}^b + v_{ep}^b + \mu^b) - \frac{2}{\tau_{pi}}z_{pi}^b - \frac{1}{\tau_{pi}^2}v_{pi}^b,\\
\frac{dv_{pe}^b}{dt} &= z_{pe}^b, & \frac{dz_{pe}^b}{dt} &= \alpha_{pe}^b g(v_{ip}^b + v_{ep}^b + \mu^b) - \frac{2}{\tau_{pe}}z_{pe}^b - \frac{1}{\tau_{pe}^2}v_{pe}^b,\\
\frac{dv_{ep}^b}{dt} &= z_{ep}^b, & \frac{dz_{ep}^b}{dt} &= \alpha_{ep}^b g(v_{pe}^b) - \frac{2}{\tau_{ep}}z_{ep}^b - \frac{1}{\tau_{ep}^2}v_{ep}^b.
\end{aligned}
\tag{13}
$$

Here, the parameters $\alpha$ and $\tau$ represent the connection strength and time constant, respectively, for local synaptic dynamics, while $w_{ab}$ quantifies inter-regional connectivity strength. Inputs from region $a$ modulate the activity of region $b$, as reflected in the term $\sum_{b=1}^N w_{ab}g(\mu^a + v_{ep}^a + v_{ip}^a)$.

The entire cortical network, comprising $N$ regions, is represented by the concatenated state vector

$$\mathbf{X} = [\mathbf{x}^1, \mathbf{x}^2, \ldots, \mathbf{x}^N]^\top. \tag{14}$$

The dynamics of the full network are captured by

$$\dot{\mathbf{X}} = f(\mathbf{X}), \tag{15}$$

where $f$ defines the system of equations mapping $\mathbb{R}^{10N}$ to $\mathbb{R}^{10N}$, as specified by Eq. (13).

This formulation allows for the modeling of both local neural dynamics and large-scale cortical interactions, providing a biologically grounded framework for simulating brain activity.

## Parameter estimation from data

The parameter estimation contains two steps. The first step involves applying the semi-analytical Kalman filter that was developed in "NeuroProcImager"[37] to infer the parameters in each neural mass model from MEG data. This method provides unbiased estimates of both state vectors and parameters by minimizing the mean square error, assuming the underlying states follow a Gaussian distribution. The inhibitory connection parameter $\alpha_{ip}$ was fixed to a constant value during the filtering process, as Xenon is known to have minimal impact on GABAergic activity[93]. This fixed value was derived from earlier studies[88]. To initialize the Kalman filter, the *a posteriori* state estimate $\hat{\xi}_0^+$ and covariance matrix $\hat{\mathbf{P}}_0^+$ at $t = 0$ were determined based on the mean and covariance of simulated data corresponding to the resting state. Recognizing that the initial filter configuration might not perfectly align with the MEG data, the first five seconds of the recordings were excluded from the analysis. This step ensured that the filter had sufficient time to stabilize and produce reliable estimates. Small constant values ($5\,\mu V$) were assigned to the model noise covariance $\mathbf{Q}$ to prevent full convergence of the filter and ensure continuous influence from new observations. Measurement noise was set to $1\,\mu V$, following the recommendations of prior research[37,89]. The semi-analytical Kalman filter was applied to each source time series to estimate time-varying regional NMM parameters ($\hat{\alpha}_{mn}, \hat{\mu}$) and state variables ($\hat{v}_{mn}, \hat{z}_{mn}$).

To determine the inter-regional connectivity between regions (i.e., $w_{ab}$), the input $\mu$ to region $b$ was computed as the summation of post-synaptic membrane potentials caused by firing rates from other regions

$$\mu^b(t) = \sum_{a=1}^{N} w_{ab} \int_0^t h_{ab}(t - t') g(v_p^a(t'))\, \mathrm{d}t', \qquad (16)$$

where the kernel $h_{ab}(t)$ is defined as

$$h_{ab}(t) = \eta(t) \frac{t}{\tau_d} \exp\left(-\frac{t}{\tau_d}\right), \qquad (17)$$

where $\tau_d$ is the decay time constant for inter-regional synaptic connections[94]. The estimation of $w_{ab}$ was carried out using multivariate linear regression on data segmented into one-second intervals. This temporal resolution is supported by neurophysiological evidence, as inter-regional interactions are known to integrate information over relatively longer time scales compared to local neuronal dynamics. Such an approach is consistent with established methodologies in the field[95–98]. This multivariate linear regression was also adopted to handle the computational demands of the whole-cortex model, which has a high dimensionality (780 state variables and 6396 parameters), making Kalman filters computationally expensive and impractical. Specifically, let $\mathbf{M}$ represent the matrix of estimated $\mu$ for all regions, $\mathbf{Y}$ the matrix with each column containing the convolution of the $v_p$ estimates with the post-synaptic response kernel, $\mathbf{W}$ the inter-regional connectivity coefficients, and $\mathbf{E}$ the intercept matrix which can explain signals from the thalamus. The linear regression model is expressed as

$$\mathbf{M} = \mathbf{Y}\mathbf{W} + \mathbf{E}. \qquad (18)$$

The maximum likelihood estimator for $\mathbf{W}$ is

$$\hat{\mathbf{W}} = (\mathbf{Y}^\top \mathbf{Y})^{-1} \mathbf{Y}^\top \mathbf{M}. \qquad (19)$$

Connectivity matrices $\mathbf{W}$ were computed for each 1-second time window, enabling the derivation of time-varying connectivity patterns across MEG recordings. This strategy effectively balances computational efficiency and estimation accuracy, avoiding the prohibitive memory and processing requirements of Kalman filtering in large-scale models.

The inference framework was validated by examining the prediction error of the Kalman filter, which followed an approximately normal distribution with a mean close to zero (Fig S2). The inter-regional connectivity estimates obtained from the multivariate regression model demonstrated a strong correlation with the observed data, achieving minimum $R^2 \geq 0.89$. This approach successfully combined the precision of Kalman filtering for local parameters with the efficiency of linear regression for inter-regional connectivity, making it a robust solution for whole-cortex modeling.

## Local stability near the fixed points for whole-cortex model

The equilibrium points of the whole-cortex model, denoted by $\mathbf{X}^*$, were determined by setting $\dot{\mathbf{X}} = 0$ in Eq. (15) and solving the resulting equations. To identify equilibrium points near the model's estimated states, the solver utilizes these estimates as initial conditions for the numerical computation of the solution. To study the behavior of the system near equilibrium points, the whole-cortex model was linearized, and the Jacobian matrix at the equilibrium point $\mathbf{X}^*$ was calculated.

The system comprises $N$ cortical regions, each represented by a neural mass model of 10 dimensions. As a result, the Jacobian matrix $\mathbf{J} \in \mathbb{R}^{10N \times 10N}$ for the entire model is structured as follows

$$\mathbf{J} = \mathrm{diag}(\mathbf{M}_1, \dots, \mathbf{M}_N) + \begin{bmatrix} & \mathbf{0}_{9,10N} & \\ \mathbf{\Omega}_{1,1} & \cdots & \mathbf{\Omega}_{N,1} \\ & \mathbf{0}_{9,10N} & \\ \mathbf{\Omega}_{1,2} & \cdots & \mathbf{\Omega}_{N,2} \\ \vdots & \ddots & \vdots \\ & \mathbf{0}_{9,10N} & \\ \mathbf{\Omega}_{1,N} & \cdots & \mathbf{\Omega}_{N,N} \end{bmatrix}, \qquad (20)$$

where $\mathbf{\Omega}_{a,b}$ captures the coupling between regions $a$ and $b$, defined as

$$\mathbf{\Omega}_{a,b} = \begin{bmatrix} w_{ba} v_1 & 0 & 0 & w_{ba} v_1 & w_{ba} v_1 & 0 & 0 & 0 & 0 & 0 \end{bmatrix}, \qquad (21)$$

with $v_1 = g'(\mu^b + v_{ip}^b + v_{ep}^b)$. The term $\mathbf{0}_{k,l}$ represents a matrix of zeros (size $k \times l$), and $\mathrm{diag}(\mathbf{M}_1, \dots, \mathbf{M}_N)$ is the block diagonal matrix with components $\mathbf{M}_a \in \mathbb{R}^{10 \times 10}$, given by

$$\begin{bmatrix} & & \mathbf{0}_{5,5} & & & & & \mathbf{I}_{5,5} & & \\ -\frac{1}{\tau_{ip}^2} & \alpha_{ip}^a g'(v_{pi}^a) & 0 & 0 & 0 & \frac{-2}{\tau_{ip}} & 0 & 0 & 0 & 0 \\ \alpha_{pi}^a v_2 & \frac{-1}{\tau_{pi}^2} & 0 & \alpha_{pi}^a v_2 & \alpha_{pi}^a v_2 & 0 & \frac{-1}{\tau_{pi}} & 0 & 0 & 0 \\ \alpha_{pe}^a v_2 & 0 & \frac{-1}{\tau_{pe}^2} & \alpha_{pe}^a v_2 & \alpha_{pe}^a v_2 & 0 & 0 & \frac{-2}{\tau_{pe}} & 0 & 0 \\ 0 & 0 & \alpha_{ep}^a g'(v_{pe}^a) & \frac{-1}{\tau_{ep}^2} & 0 & 0 & 0 & 0 & \frac{-2}{\tau_{ep}} & 0 \\ 0 & 0 & 0 & 0 & \frac{-1}{\tau_d^2} & 0 & 0 & 0 & 0 & \frac{-2}{\tau_d} \end{bmatrix}. \qquad (22)$$

Here $\mathbf{I}_{k,k}$ is the identity matrix of size $k \times k$ and $v_2 = g'(\mu^a + v_{ip}^a + v_{ep}^a)$.

To assess linear stability of the model, the eigenvalues $\lambda_k$ of $\mathbf{J}$ were computed by performing singular value decomposition

$$\mathbf{J} = U \Lambda U^\top. \qquad (23)$$

Here $\Lambda$ is a diagonal matrix containing the eigenvalues and $U$ is an orthonormal matrix containing eigenvectors (i.e., eigenmodes). The real part of each eigenvalue determines stability along the corresponding eigenvector. A negative real part indicates stability, meaning that small perturbations along the eigenvector decay over time, causing the system to return to equilibrium. Conversely, a positive real part implies instability, where small perturbations grow, driving the system away from equilibrium. A zero real part indicates neutral stability. In conclusion, the system is unstable even there is at least one eigenvalue with positive real part, while it is stable when all eigenvalues have negative real parts.

The count of unstable eigenvalues reflects the system's level of instability, indicating the number of eigenvector-defined directions in which the system is prone to diverge. In this study, the number of unstable eigenvalues was used to track shifts in cortical stability. In summary, linear

stability analysis provides critical insights into a system's ability to withstand small disturbances. When applied to the study of loss of consciousness, this method helps elucidate how cortical dynamics evolve between stable and unstable states, offering a framework to understand the mechanisms driving transitions into unconscious states and relate NCCs to consciousness levels.

## In-degree and out-degree centrality

In-degree centrality in the inter-regional connectivity matrix refers to the measure of the total weight of incoming connections that a brain region receives from other regions. Out-degree centrality represents the measure of the total weight of outgoing connections originating from a brain region. These centrality measures help identify regions that play significant roles in receiving or transmitting weighted connections, highlighting their importance in information integration and communication within the brain network.

## Symbolic mutual information

It is a non-directed functional measure of global information sharing between points of interest. Prior EEG studies have shown its superior discriminatory power across altered states of consciousness such as vegetative and minimally conscious states when compared to other connectivity metrics, including the Phase Lag Index[47,48]. It quantifies the joint probability between pairs of voxels transformed into discrete symbols of m time samples. The measure was computed on 3s windows, with a temporal resolution of 1ms. Following the computation of the probability of each symbol in each signal, the joint probabilities between channel pairs were extracted. The joint probability matrix was multiplied by binary weights to reduce spurious correlations between signals. The weight was set to 0 for identical symbol pairs and opposite symbols (e.g., [3 1 2] and [1 3 2]), which could be generated via a common source, or opposite sides of a single dipole, respectively.

The computation is summarized as follows:

$$\mathrm{wSMI}(X, Y) = \frac{1}{\log m!} \sum_{x \in X} \sum_{y \in Y} w(x, y)\, p(x, y) \log\left(\frac{p(x, y)}{p(x)\, p(y)}\right)$$

with

$$w(x, y) = \begin{cases} 0, & x = y \quad \text{or} \quad x = m - y + 1, \\ 1, & \text{otherwise}. \end{cases}$$

Here, $x$ and $y$ represent all symbols of length $m$ in channel signals $X$ and $Y$, $p(x, y)$ is the joint probability of symbol $x$ in $X$ co-occurring with symbol $y$ in $Y$, and $w(x, y)$ denotes the weight matrix between symbols.

## Permutation tests

The foundational principles of permutation tests are extensively explained in the works of Nichols and Holmes[99,100]. The procedure for performing a permutation test consists of the following steps: Begin with several observations under two separate conditions. Let $N$ represent the total number of potential label permutations, and $t_i$ indicate the test statistic corresponding to the $i$-th permutation. The permutation distribution is then formed by aggregating all $t_i$ values across every possible permutation. Define the observed test statistic from the original experimental labels as $T$. Assuming the null hypothesis, each $t_i$ has an equal likelihood of occurrence, enabling the computation of the $p$ value by determining the fraction of the permutation distribution that is at least as extreme as $T$. If this $p$ value falls below the significance level $\alpha$, the null hypothesis $H_0$ is rejected at the $\alpha$ level.

As an illustration, a correlation analysis was performed to examine the relationship between responsiveness and intra-prefrontal connection strength, utilizing data from 15 participants. For each subject, a correlation coefficient was calculated, resulting in 15 individual values. The statistic of interest was defined as the average of these coefficients. Under the null hypothesis, which assumes no correlation between responsiveness and intra-prefrontal connection strength, relabeling was executed by randomly

assigning a factor of either 1 or –1 to the correlation coefficients. The statistic was then recomputed for each relabeling. This procedure was repeated extensively (100,000 times in this case) to construct the permutation distribution of the statistic. The proportion of the distribution with values as extreme as or more extreme than $T$ was subsequently calculated to obtain the $p$ value. Finally, the $p$ value was compared against the significance threshold $\alpha$ to determine whether to reject the null hypothesis $H_0$.

## Multiple comparisons permutation tests

In our study, the challenge of multiple comparisons arises when simultaneously testing hypotheses across numerous voxels or connections in the human brain for multiple subjects. Specifically, statistical calculations are performed using data from each subject for every voxel or connection, generating a $p$ value corresponding to the null hypothesis for each voxel or connection. Controlling Type I errors collectively is essential to ensure that the probability of mistakenly identifying any voxel or connection as significant remains below the specified threshold $\alpha$. To address this, we employ a testing procedure developed by Nichols and Holmes, which effectively manages Type I errors by providing adjusted $p$ values. Readers are encouraged to read[99].

To ensure the validity of an omnibus test under the assumption that all null hypotheses across voxels or connections are true, label exchangeability under the omnibus null hypothesis must be preserved. This condition is crucial for any summary statistic derived from voxel- or connection-level analyses, such as the maximum statistic. The critical threshold for significance is obtained by evaluating the distribution of the maximum statistic across the entire volume of interest. Instead of calculating permutation distributions for individual voxels or connections, a permutation-based approach is used to generate the distribution of the maximal statistic. A voxel or connection is deemed significant if its test statistic surpasses the $j + 1$-th largest value in this permutation-derived distribution, where $j = [\alpha N]$ and $N$ is the total number of permutations. The corrected $p$ value for each voxel or connection is determined by calculating the fraction of the permutation distribution of maximum statistics that are equal to or greater than the observed statistic. This approach provides a practical and robust means of adjusting for multiple comparisons across the entire dataset, ensuring control of the family-wise error rate.

For instance, In the section titled Correlation of consciousness and inter-regional connectivity, we identify brain structures where the time-varying in-degree centrality significantly correlates with responsiveness. We conduct a correlation analysis between in-degree centrality and responsiveness for each brain structure and subject, resulting in an $n \times m$ correlation coefficient matrix. Here $n$ denotes the number of subjects and $m$ represents the number of brain structures. The t-statistic for each brain structure is computed using the correlation coefficients from the $n$ subjects, yielding a $1 \times m$ t-statistic vector. This vector summarizes the correlation information across subjects.

The null hypothesis for each brain structure assumes no correlation between time-varying in-degree centrality and responsiveness. To test this hypothesis, we use a permutation approach by randomly permuting the correlation coefficient matrix while maintaining the null hypothesis. Each element is randomly multiplied by either 1 or $-1$. The t-statistic vector is recalculated for the permuted matrix, and the maximum and minimum values are recorded. Repeating this permutation process $N$ times generates permutation distributions for the maxima and minima. For each brain structure, the null is rejected if the observed t-statistic exceeds the $(100 - \alpha/2)$th percentile of the maximum value distribution or falls below the $(\alpha/2)$th percentile of the minimum value distribution.

## Statistics and reproducibility

All statistical analyses were performed at the participant level ($n = 15$ independent participants), treating each participant as a biological replicate; time-resolved windows within a participant were handled as repeated measures and not independent replicates. Group-level inferences used nonparametric permutation tests with family-wise error controlled via a

maximum-statistic procedure for connection-/voxel-level maps; where multiple comparisons were performed, we controlled the false discovery rate using the Benjamini-Hochberg procedure at $\alpha = 0.05$. Within-subject associations (e.g., cortical stability versus consciousness or anesthetic level) were computed per participant and then summarized across the cohort. Any exclusions are reported alongside the relevant analysis (e.g., two participants excluded from a specific subanalysis due to incomplete equilibration). Time-varying connectivity and stability metrics were estimated in successive 1s windows; weighted symbolic mutual information (wSMI) was computed in 3s windows at 1ms resolution. Code to reproduce the analysis is available (NeuroProcImager-Pro).

## Ethical statement
Informed consent was obtained from all participants prior to data collection. The study protocol was approved by the Alfred Hospital Human Research Ethics Committee (Alfred Health, Melbourne, Australia; approval 260/12). All ethical regulations relevant to human research participants were followed.

## Reporting summary
Further information on research design is available in the Nature Portfolio Reporting Summary linked to this article.

## Data availability
The source-level MEG data can be requested to the Authors. All intermediate source data for making graphs and charts in this paper is available on Figshare (https://figshare.com/account/home#/projects/259940)[101–105].

## Code availability
The MATLAB code for the whole-cortex model, parameter estimation, and dynamic cortical stability analysis is available on GitHub NeuroProcImager-Pro (https://github.com/yundumbledore/NeuroProcImager-Pro).

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

## Acknowledgements
We thank MASSIVE for computational resources. This work was supported by Australian Research Council Discovery Project Grants (DP200102600, DP210100045).

## Author contributions
All authors wrote the paper. Conceptualization involved Y.Z., N.T., and L.K. Data collection involved A.P., W.W., D.L., and L.K. Development of the whole-cortex model framework involved Y.Z., M.B., Y.L., P.K., and L.K. Computations and experiments involved Y.Z., V.D., and L.K.

## Competing interests
The authors declare no competing interests.
