## [Transparent Peer Review file · Communications Biology]

Cortical connectivity, local dynamics and stability correlates of global conscious states

Corresponding Author: Dr Levin Kuhlmann

Version 0:

Reviewer comments:

Reviewer #2

(Remarks to the Author)

Zhao and colleagues investigate, in 15 MEG recordings of healthy participants under Xenon-induced anesthesia, how the level of consciousness correlates with different neuronal features that arise from whole-brain computational models. They built individual whole-brain cortex models using a two-step optimization procedure based on a well-known neural mass model, named Jensen and Rit, to define the local dynamics of each brain region. As a result of the fitting process, they obtained model parameters that change over time, as well as the effective connectivity between brain regions for each participant. They then correlated these model outputs with an estimation of consciousness, defined as the performance in an auditory continuous task.

I found that the manuscript addresses an important question and proposes an interesting methodology to tackle it. The paper is well-written and clearly presented. However, I suggest several changes to improve the robustness of the results and enhance the clarity of the manuscript.

Major Comments:

1. The first sentence of the abstract states that consciousness emerges as a neural process, which is a claim that could be debated depending on the perspective. I suggest revising this statement to soften its tone.
2. In the introduction, the authors mention that model measures will be correlated with task performance (line 102), but the task itself is not clearly defined. It would be helpful to provide a more detailed explanation in this section, considering that the task is a central aspect of the manuscript.
3. The link between task performance as a proxy for the level of consciousness should be further explored and justified. For instance, it would be valuable if the authors demonstrated how this measure correlates with Xenon concentration and possibly with other measures extracted directly from MEG data that are typically used as biomarkers of consciousness (e.g., this study, among others). Additionally, the authors should examine whether the results obtained in Sections 2.1, 2.2, and 2.3 remain consistent when correlating with Xenon concentration instead of the level of consciousness. If not, the interpretation of these findings should be clarified.
4. Figure 1:
 - o Panel A: The diagram of the model should be made clearer, as some arrows cross over text boxes, making it difficult to follow.
 - o Panels B to F: These panels are interesting but represent data from only one participant. Showing an averaged result across the 15 participants or a way to summarize the full experiment's findings could be more informative.
5. The manuscript uses both "effective connectivity" and "inter-regional connectivity" in different sections. Since both terms seem to refer to the same inferred connectivity from the model, I suggest unifying the terminology for consistency.
6. Given that the manuscript heavily relies on modeling results, I recommend including an evaluation of the model's performance in the main ms. Specifically, how well does the model fit the data for each participant and each brain region? Additionally, the authors should reference studies demonstrating that the level of model fitting is sufficient to represent brain dynamics and compared with other studies using whole-brain models.
7. It would be great if the results showed in table 2 are showing in a figure as Fig 2C using networks as nodes of the circular graph to be more accessible and clearer.
8. In fig4 it would be great if the authors include a panel summarizing the results that are included in table 4, like a plot showing all the correlations.
9. In section 2.5 the authors investigate the stability of the system based on the Jacobian of the matrix. Considering that many whole-brain models works demonstrated that the optimal working point is just below a hopf bifurcation I was wondering if the authors could include some analysis or/and discussion how this stability is related with the bifurcations of

the system and if there is changes in the bifurcation landscape in time with the level of consciousness or not.

Reviewer #3

(Remarks to the Author)

In this study, the authors aim at clarifying the debate surrounding the localization of the NCC: are they rather posterior (as postulated by IIT) or frontal (as postulated by GWT)? To do so, the authors use a previously published dataset in combination with a whole-brain NMM, and attempt at linking model parameters associated with physiological processes. This question is key in the field of consciousness studies, and will attract a wide readership. The paper is very well-written, and provides generally compelling results that are in favor of IIT, bringing long awaited data to (at least partially) settle this debate. I have a few concerns however that I would like to see addressed in a revision:

- the authors only include one type of inhibitory interneurons, however it is probably important to include slow (SST) and fast (PV) interneurons to account for the various types of oscillations involved in consciousness-associated activity.
- the robustness of the conclusions with respect to the modeling choices should be discussed.
- the authors should emphasize why the use of the model is instrumental in obtaining their results, i.e. why those results could not have been obtained through other means (e.g., Granger causality analyses).
- is the use of the number of unstable eigenmodes to assess cortical stability standard? If yes, appropriate references should be cited, and if not, this should be justified further, and why other potential metrics could not be used.
- the Discussion could be fleshed out a bit more, by discussing for example previous works using comparable whole-brain model approaches, and how the present model and associated analyses stands out.

Version 1:

Reviewer comments:

Reviewer #2

(Remarks to the Author)

I appreciate the efforts made by the authors in addressing all my comments. In my opinion, the current version of the ms is suitable to be published in this journal.

Dear Editor and Reviewers,

Thank you for your thoughtful and constructive feedback on our manuscript entitled “Cortical connectivity, local dynamics and stability correlates of global conscious states” (manuscript ID: COMMSBIO-25-0106-T). We appreciate the time and effort you invested in evaluating our work. We have carefully considered each of your comments and have revised the manuscript accordingly.

Below, we provide a point-by-point response to all suggestions and concerns. Please note that the figures included in this response letter retain the same numbering as in the manuscript. All reference citations in black in this letter correspond to its own bibliography, and any reference numbers shown in blue text have been copied directly from the manuscript to indicate their original placement in the manuscript’s reference list.

Comments from Reviewer 1:

1. The first sentence of the abstract states that consciousness emerges as a neural process, which is a claim that could be debated depending on the perspective. I suggest revising this statement to soften its tone.

My response: Thank you for pointing out that the original statement regarding consciousness emerging from neural processes might be perceived as overly definitive. In response to your suggestion, we have revised the first sentence of the abstract to a softer tone:

“Waking levels of human consciousness are known to be supported by the integrity of complex structures and processes in the brain, yet how they are exactly regulated by neurobiological mechanisms remains uncertain.”

2. In the introduction, the authors mention that model measures will be correlated with task performance (line 102), but the task itself is not clearly defined. It would be helpful to provide a more detailed explanation in this section, considering that the task is a central aspect of the manuscript.

My response: Thank you for highlighting the need for a detailed description of the task in the Introduction. In response, we have expanded our discussion to clarify how the auditory continuous performance task (ACPT) is administered and why it serves as a surrogate measure for global conscious states. Specifically, we now explain that participants receive tones of different frequencies through headphones and are asked to respond via button presses, enabling us to assess their level of alertness or awareness during sedation. This information has been incorporated into the Introduction to underscore how ACPT accuracy is used as a proxy for consciousness level, and to clarify its central role in correlating with our model measures. We hope this revision (see below) provides a complete understanding of the task and its importance to the study.

“Specifically, participants’ behavioural performance was measured using an auditory continuous performance task (ACPT). The ACPT is designed to assess vigilance and responsiveness: participants hear tones of different frequencies—1 or 3 kHz—through headphones and must press one of two corresponding buttons to indicate which tone they heard. The accuracy and speed of their responses provide a quantitative measure of their alertness (or level of consciousness) as they progress through varying levels of sedation. Because it relies on sustained attention and working memory, the ACPT accuracy is sensitive to changes in consciousness and can thus be leveraged as a reliable surrogate measure for global conscious states. By correlating our model estimates with performance on the ACPT, we aim to capture how changes in underlying neural mechanisms reflect behavioural responsiveness and, by extension, global consciousness.”

3. The link between task performance as a proxy for the level of consciousness should be further explored and justified. For instance, it would be valuable if the authors demonstrated how this measure correlates with Xenon concentration and possibly with other measures extracted directly from MEG data that are typically used as biomarkers of consciousness (e.g., this study, among others).

Additionally, the authors should examine whether the results obtained in Sections 2.1, 2.2, and 2.3 remain consistent when correlating with Xenon concentration instead of the level of consciousness. If not, the interpretation of these findings should be clarified.

My response: We appreciate the reviewer’s suggestion to further explore and justify the use of task performance as a proxy for consciousness levels. We have shown in Section 2.5 a detailed correlation analysis between our “Consciousness level” measure and administered Xenon concentration levels. The results of this analysis are presented in Table 4 (Columns 5 and 6), where we report Pearson’s correlation coefficients alongside their associated p-values, demonstrating a negative association between Xenon concentration levels and task performance. Additionally, Figure 5b’s third boxplot panel visualizes the distribution of the correlation coefficients across all subjects. These findings confirm that higher levels of Xenon anaesthesia correspond to lower task performance scores (i.e., “Consciousness level”).

In addition, we now present a comparison between ACPT accuracy and weighted Symbolic Mutual Information (wSMI) [1, 2], a recognised functional connectivity index of conscious processing. Prior EEG studies have shown its superior discriminatory power across altered states of consciousness such as vegetative and minimally conscious states when compared to other connectivity metrics, including the Phase Lag Index. Using the MEG data, we computed wSMI for four brain networks: whole brain, posterior parietal cortex, prefrontal cortex, and the posterior parietal to prefrontal connections, and tracked these metrics alongside ACPT accuracy throughout the Xenon-sedation protocol. Across participants, ACPT accuracy showed statistically significant positive correlations with whole brain wSMI, posterior parietal wSMI, and the posterior parietal to prefrontal wSMI for most subjects, whereas the correlation with the prefrontal wSMI was not significant. These patterns are illustrated for a representative individual (See the figure below) and are reported at the group level (See the right side of the table below) and are consistent with our inferred long-range NMM connectivity changes observed in the exact same brain networks.

Figure S1 Time-resolved relationship between behavioural responsiveness and electrophysiological information sharing during Xenon sedation. Blue traces depict Auditory Choice Performance Task (ACPT) accuracy (left y-axis) and orange traces show weighted Symbolic Mutual Information (wSMI; right y-axis) extracted from magnetoencephalography (MEG) in a representative participant. wSMI was computed in brain networks for the (left-to-right) whole brain, posterior parietal cortex, posterior parietal to prefrontal cortex, and prefrontal cortex. In each panel the Pearson correlation between ACPT accuracy and the corresponding wSMI time-series is indicated in the title.

Correlating connection strength to ACPT accuracy		Correlating wSMI to ACPT accuracy	
Network(s)	p	Network(s)	p
Posterior parietal	0.0475	Posterior parietal	0.0212
Posterior parietal→Prefrontal	0.0495	Posterior parietal→Prefrontal	0.0397
		Whole brain	0.0191

Table 3 Networks whose connection strength or wSMI measures showed a significant correlation with consciousness levels (as measured by ACPT accuracy). Arrows indicate the connections between two networks. Permutation tests were applied to find group-level p-values (see “Methods”).

We have added a paragraph about this new measure to Section 2.3.

“To complement our inter-regional connectivity findings and provide a functional perspective on information exchange especially since weighted symbolic mutual information (wSMI) is a validated biomarker of consciousness levels (e.g., sensitive to distributed, time-synchronized signaling across neural populations) [47, 48], we computed wSMI for four networks: whole brain, posterior parietal cortex, prefrontal cortex, and the posterior parietal to prefrontal connections and correlated these measures with ACPT accuracy (i.e., consciousness levels). Fig S1 shows the time-resolved relationship between ACPT accuracy and wSMI for an example subject. Having identified that posterior parietal network and posterior parietal to prefrontal connection strength were associated with ACPT accuracy, we sought to determine if functional integration in the same regions was correlated with ACPT accuracy. Importantly, we show that (right side of Table 3) greater wSMI within the posterior parietal network, the posterior parietal to prefrontal network and across the whole brain, all correlate with ACPT accuracy. Because wSMI quantifies the extent of functional integration, these findings suggest that subjects with more coherent, functionally integrated activity particularly in posterior parietal regions tend to have higher levels of consciousness. Crucially, this biomarker converges with our earlier inter-regional connectivity results: the intra-network connection strength of the posterior parietal network and its projections to prefrontal network were significantly correlated with ACPT accuracy. In other words, both inter-regional connectivity and functional measures single out posterior parietal and its communication with prefrontal cortex as essential for modulating conscious states.”

For the wSMI description, we added a subsection to the Methods.

“It is a non-directed functional connectivity measure of global information sharing between points of interest. Prior EEG studies have shown its superior discriminatory power across altered states of consciousness such as vegetative and minimally conscious states when compared to other connectivity metrics, including the Phase Lag Index [47, 48]. It quantifies the joint probability between pairs of voxels transformed into discrete symbols of m time samples. The measure was computed on 3s windows, with a temporal resolution of 1ms. Following the computation of the probability of each symbol in each signal, the joint probabilities between channel pairs were extracted. The joint probability matrix was multiplied by binary weights to reduce spurious correlations between signals. The weight was set to 0 for identical symbol pairs and opposite symbols (e.g., [3 1 2] and [1 3 2]), which could be generated via a common source, or opposite sides of a single dipole, respectively.

The computation is summarized as follows,

$$wSMI(X, Y) = \frac{1}{\log m!} \sum_{x \in X} \sum_{y \in Y} w(x, y) p(x, y) \log \left(\frac{p(x, y)}{p(x)p(y)} \right)$$

$$w(x, y) = \begin{cases} 0, & x = y \text{ or } x = m - y + 1, \text{ otherwise} \end{cases}$$

where x and y represent all symbols of length m in channel signals X and Y , $p(x, y)$ is the joint probability of the co-occurrence of symbol x in signal X and symbol y in signal Y . $w(x, y)$ denotes the weight matrix between symbols.”

We agree that examining correlations between the model parameter estimates and Xenon concentration levels, rather than only level of consciousness, could provide additional insights. However, performing this analysis lies beyond the scope of the study. We therefore acknowledge in the Discussion section that we have not yet assessed the correlation between parameter estimates and Xenon concentration level and propose to undertake this evaluation in future work. Below is a statement we added to the Discussion section.

“We also recognize that our analysis focused on how neurophysiological variables relate to the level of consciousness and did not explicitly test their relationship with Xenon concentration levels. We therefore acknowledge this as a study limitation and plan to explore dose-dependent effects in future work by acquiring higher-resolution Xenon measurements and assessing how parameter estimates and state trajectories scale with inhaled concentration.”

4. Figure 1:

- o Panel A: The diagram of the model should be made clearer, as some arrows cross over text boxes, making it difficult to follow.
- o Panels B to F: These panels are interesting but represent data from only one participant. Showing an averaged result across the 15 participants or a way to summarize the full experiment's findings could be more informative.

My response: o Panel A: Thank you for this suggestion. We have revised Panel A (see below) so that arrows have been repositioned and no longer cross any text boxes, improving the clarity of the diagram.

Figure 2 Revised Panel A.

o Panels B to F: Thank you for the suggestion. Panel B–F are intended to illustrate a single, representative example of our full analysis framework—beginning with the 78 source-point time series, ACPT accuracy (i.e., consciousness levels) and Xenon

concentration levels from one subject, through to inter-regional connectivity and local synaptic and field-potential estimates. Therefore, we have retained the illustrative example. But we created a new panel in Figure 1 (shown below) to summarize the parameter estimation results across all subjects. State variable estimates have been omitted here because their oscillatory nature makes their mean values uninformative.

Figure 1g Time courses of averaged local and inter-regional connection strength across all subjects, with the 95% confidence interval shown as gray shaded bands.

Moreover, we added the following description in Section 2.

“Figure 1g illustrates the group-level dynamics of both local and inter-regional coupling under Xenon-induced anaesthesia across all subjects. During reduced consciousness, the mean strengths of the pyramidal-to-inhibitory population connection α_{pi} and the pyramidal-to-excitatory connection α_{pe} both exhibit a clear downward shift from baseline, whereas the excitatory-to-pyramidal connection strength α_{ep} remains statistically stable. The averaged inter-regional connectivity strength declines significantly during low-consciousness periods and then rebounds as consciousness is regained. Together, these group-level results confirm that reduced levels of consciousness are accompanied by a selective weakening of local excitatory connections and by an overall reduction in long-range cortical coupling.”

- The manuscript uses both "effective connectivity" and "inter-regional connectivity" in different sections. Since both terms seem to refer to the same inferred connectivity from the model, I suggest unifying the terminology for consistency.

My response: We thank the reviewer for pointing out this inconsistency. We have now unified the terminology throughout the manuscript, replacing every instance of “effective connectivity” with “inter-regional connectivity” to ensure consistency.

- Given that the manuscript heavily relies on modeling results, I recommend including an evaluation of the model's performance in the main ms. Specifically, how well does the model fit the data for each participant and each brain region? Additionally, the authors should reference studies demonstrating that the level of model fitting is sufficient to represent brain dynamics and compared with other studies using whole-brain models.

My response: We thank the reviewer for highlighting the importance of quantitatively validating our model's ability to track empirical MEG signals. At the core of our fitting

procedure is an analytic Kalman filter (AKF), whose accuracy and convergence we evaluated in Zhao et al [3]. In Supplementary Figures 1–4 of that paper, we demonstrate that the AKF converges to true parameter values with errors under 6%, tracks all model state variables with errors below 2.3%, and achieves normalized signal-tracking RMSE under 2.5% across a broad parameter space—while running twice as fast as a comparable unscented Kalman filter.

When applied to real MEG source-space time series, the AKF’s pointwise prediction error is narrowly distributed around zero (see Fig S4) and preserves key oscillatory dynamics. To illustrate this in the present manuscript, we have added a new figure (shown below) displaying boxplots of the RMSE—computed as the difference between the AKF prediction and the raw MEG signal—for each of our fifteen subjects. Importantly, these RMSE values amount to $\leq 10\%$ of each signal’s dynamic range, confirming that the AKF is an optimal recursive estimator whose effect is equivalent to adaptive low pass filtering—to attenuate high frequency noise while preserving slower dynamics.

Figure S2 Root mean squared error between measured and predicted MEG signals for all subjects. Box plots show median (red bar), 1st and 3rd quartiles of the RMSE values for each subject. Median of the RMSE distributions ranged from 0.7 to 3.1, and the amplitude of the measured MEG signal ranges from approximately 15 to 32, reflecting errors below 10% for real data.

Moreover, the same modelling and Kalman filter framework have been applied to study different brain conditions, including epileptic seizure dynamics [4, 5], anaesthetized states [6], and resting-state alpha-rhythm modulation [3]. In each application, the predicted signals are shown to be closely tracking the raw measurement, demonstrating that our framework is sufficiently robust and generalizable to represent core aspects of brain dynamics.

We have accordingly added a paragraph in Section 2 to highlight the model fitting issue and discuss the Kalman filter is able to track the raw signals and the estimated model parameters are precise enough to represent the underlying brain dynamics.

“Model fitting in our framework is performed using an analytic Kalman filter whose performance was comprehensively evaluated in another paper [37], and those results demonstrate that the filter can accurately track raw source-space MEG signals while recovering underlying model parameters with high precision. The study showed that

the recovered parameters closely match their ground-truth values with errors below 6%, indicating that the fitted parameter estimates are sufficiently precise to serve as biophysically meaningful representations of the underlying cortical processes. The same modelling framework has been applied to study epileptic seizures [42] and resting state alpha rhythm [37]. In this study, we present a subject-wise RMSE boxplots in Fig S2 showing that prediction errors remain below 10% of each signal's dynamic range across all fifteen participants, confirming that the Kalman filter effectively attenuates high-frequency noise yet faithfully preserves the slower oscillatory components that carry the core brain dynamics."

7. It would be great if the results showed in table 2 are showing in a figure as Fig 2C using networks as nodes of the circular graph to be more accessible and clearer.

My response: Thank you for the suggestion to include a network visualization. In response, we have added a new diagram (Figure 3) in Section 2.2 that displays all statistically significant intra- and inter-network connections whose strengths correlate with consciousness levels. Posterior parietal and prefrontal regions have been omitted from this figure because they do not constitute the core functional networks examined here and are addressed separately in Section 2.3.

Specifically, we added below Figure 3 to section 2.2 with the following statement.

"Correlation analyses between consciousness levels and both intra- and inter-network connection strengths were conducted, with the results summarized in Fig 3 (see Table 2 for detailed statistics)."

Figure 3 Significant intra- and inter-network connections. Nodes represent functional networks, with filled nodes indicating those whose intra-network connectivity strength correlates significantly with consciousness level; unfilled nodes denote non-significant intra-network correlations. Edges encode directed inter-network connections: each link is drawn as a smooth arc whose width reflects the strength of significance, and whose colour runs from deep indigo at the origin (source of influence) to light red at the termination (target of influence).

8. In fig4 it would be great if the authors include a panel summarizing the results that are included in table 4, like a plot showing all the correlations.

My response: Thank you for the helpful suggestion. We have now augmented Fig 5 (used to be Fig 4) with a new bottom panel (panel b) that summarizes the within-subject correlations reported in Table 4. Specifically, panel b presents boxplots (with overlaid individual data points) of Pearson correlation coefficient values for (i) cortical stability vs consciousness level, (ii) cortical stability vs Xenon concentration level, and (iii) consciousness level vs Xenon concentration level. This addition allows readers to grasp the distribution and significance of these correlations at a glance. The figure caption and main text (in Section 2.5) have been updated to describe this new panel.

Figure 5 b Boxplots summarizing Pearson correlation coefficients across subjects for three pairings: cortical stability vs consciousness level; cortical stability vs Xenon concentration level; and consciousness level vs Xenon concentration level. Boxes span the interquartile range (median line), whiskers denote the full range, and individual subject values are overlaid as jittered points.

We added the following paragraph in section 2.5 to describe Figure 5b.

“Fig 5b summarizes the within-subject Pearson correlation coefficients for the three variable pairings across all 15 subjects. The boxplot on the left shows that correlations between cortical stability and consciousness level are generally positive (median ≈ 0.4 , IQR ≈ 0.5), indicating a strong coupling of instability with loss of consciousness. By contrast, the middle boxplot (cortical stability vs. Xenon concentration) is centered slightly below zero, and the right boxplot (consciousness level vs. Xenon concentration) exhibits consistently negative correlations (median ≈ -0.2 , IQR ≈ 0.4). Together, these results confirm that cortical stability tracks changes in consciousness more closely than it does Xenon concentration, and that high Xenon concentration level is accompanied by loss of consciousness.”

9. In section 2.5 the authors investigate the stability of the system based on the Jacobian of the matrix. Considering that many whole-brain models works demonstrated that the optimal working point is just below a hopf bifurcation I was wondering if the authors could include some analysis or/and discussion how this stability is related with the bifurcations of the system and if there is changes in the bifurcation landscape in time with the level of consciousness or not.

My response: Thank you for highlighting the value of framing our stability results within a bifurcation context. In the revised manuscript we have inserted a new discussion paragraph at the end of Section 2.5 explaining that each change in the number of Jacobian eigenvalues with positive real part reflects motion toward or away from a Hopf surface, thereby directly linking our time-resolved stability metric to the Hopf-bifurcation framework widely used in whole-brain modelling. We note that the systematic decline in unstable modes during Xenon-induced unresponsiveness and their recovery on emergence implies that the effective control parameters drift deeper

into—and subsequently back out of—the linearly stable half-plane, constituting state-dependent excursions in the bifurcation landscape. Because a full bifurcation analysis is infeasible with hundreds of region-specific parameters, we briefly outline how future work could project the high-dimensional trajectory onto one or two physiologically interpretable variables (e.g., mean excitatory gain or global coupling) to visualise the Hopf manifold directly. Below is the new discussion paragraph.

“In Section 2.5, we quantified cortical stability by counting, at every 1-s window, how many eigenvalues of the time-resolved whole-cortex Jacobian had positive real part. In canonical dynamical systems terms, a change in the sign of the real part of a complex-conjugate pair of eigenvalues marks a (super- or sub-critical) Hopf bifurcation, beyond which a small amplitude limit cycle is born. A large body of whole-brain modelling work has argued that resting brain dynamics operate in the subcritical regime, i.e., just below this Hopf point, because this maximizes metastability, information transfer capability and susceptibility to perturbations [57–59]. Our results are consistent with that view. When participants were behaviourally responsive the cortex showed a moderate number of unstable eigenmodes, indicating that many regions sit close to—yet just on the unstable side of—the Hopf boundary. As consciousness waned under Xenon, the number of unstable modes fell, implying that the effective control parameters (e.g. local excitatory gain) moved the system deeper into the linearly stable half-plane and therefore farther from the Hopf surface. Conversely, during recovery the eigenvalues drifted back towards the imaginary axis and the count of positive-real modes rose, representing a return toward the optimal, near-critical working point. In other words, the “bifurcation landscape” itself appears to shift over time with arousal level, rather than remaining fixed while the brain merely “chooses” different attractors within it. A bifurcation analysis with our model is intractable because there are too many free parameters (i.e., 78 local models × 3 excitatory connection strengths + 78 × 78 inter-regional connection strengths), but the temporally resolved eigenvalue approach used here still captures when the system crosses local stability boundaries. Future work could project the high-dimensional parameter trajectory onto one or two physiologically interpretable summary variables, such as the mean excitatory synaptic gain or a global coupling constant, and then perform classic bifurcation analysis to visualize the Hopf surface [60]. Nevertheless, the present results can link the stability metric to the Hopf framework and show that loss (and recovery) of consciousness corresponds to systematic excursions in this bifurcation landscape.”

Comments from Reviewer 2:

1. the authors only include one type of inhibitory interneurons, however it is probably important to include slow (SST) and fast (PV) interneurons to account for the various types of oscillations involved in consciousness-associated activity.

My response: We agree that interneuron diversity is fundamental to the micro-circuitry that shapes cortical rhythms, and that separating parvalbumin-positive (PV; fast) and somatostatin-positive (SST; slow) cells would allow a richer exploration of frequency-specific mechanisms. At the mesoscopic scale probed by source-reconstructed MEG, however, the signals mainly reflect the net postsynaptic currents of large neuronal ensembles. A tradition of neural-mass and whole-brain models therefore represents inhibition with a single composite population whose gain and time constant approximate the weighted average of multiple interneuron classes (e.g. the canonical Jansen-Rit model [7], which has reproduced alpha, beta and even gamma rhythms with only one inhibitory pool [8, 9]). Many computational frameworks that have been used to study consciousness, wakefulness and anaesthesia retain this simple assumption to avoid over-parameterisation [6, 10-12]. Moreover, Xenon is a putative NMDA receptor antagonist, as such its predominant mode of action is on excitatory rather than inhibitory synapses [13].

We therefore opted for the simple single-inhibitory-population formulation in the present study, which is sufficient to reproduce the large-scale connectivity changes and stability shifts that correlate with behavioural responsiveness. In Supplementary Figures 1 and 2, we show that the model can track the raw MEG timeseries with the prediction error centred at 0, and the root mean squared error between measured and predicted MEG signals remains below 10% of each signal's dynamic range. However, we explicitly acknowledged this limitation in the manuscript and look to implement two inhibitory neuronal populations in the future improvement and conduct a comprehensive study about it including studying the identifiability of the model and test the model with human subjects' data for different brain conditions.

Below is a new paragraph added to the first paragraph of the Discussion section.

"We are aware of neural mass models that split inhibition into fast parvalbumin (PV) and slow somatostatin (SST) pools that would allow a richer exploration of frequency-specific mechanisms [50]. However, many frameworks that probe wakefulness, anaesthesia and disorders of consciousness keep a single composite inhibitory population [51–54]. Adopting this simple arrangement keeps the parameter space compact and ensures that the parameters we estimate from non-invasive MEG remain identifiable while still capturing the large-scale stability shifts that accompany conscious state changes. Moreover, Xenon is a putative NMDA receptor antagonist, as such its predominant mode of action is on excitatory rather than inhibitory synapses. This makes it less necessary to include additional inhibitory populations. In future work the model could be explicitly extended to include PV and SST populations and test whether this added detail improves performance across different brain conditions and other GABA receptor agonist-based anaesthetics."

2. the robustness of the conclusions with respect to the modeling choices should be discussed.

My response: Thank you for drawing attention to the need for a discussion of how our conclusions depend on specific modelling choices. In response, we have added a dedicated paragraph (see below) in the Discussion section.

“The biological inferences reported here are not an artefact of the particular “starter” parametrisation we adopted but instead withstand several layers of quantitative and conceptual analyses. First, the analytic Kalman filter that anchors the pipeline tracks MEG time series with subject-wise RMSEs that sit within 10% of each signal’s dynamic range (Fig S2), while the subsequent multivariate regression that estimates the inter-regional weights explains 89% of the variance of the empirical inputs in every 1-s window. Second, the same Kalman-filter and NMM architecture reliably reproduces resting alpha activity in healthy cortex [37] and captures the rapid transition from interictal to ictal phase in focal epilepsy [42]. The underlying Jansen–Rit family of models has long been used to generate EEG dynamics observed in vivo [49], confirming the model is sufficiently expressive to capture diverse cortical dynamics, and provides a reasonable trade-off between biological realism and model complexity to enable efficient and less degenerate model inference. Third, all statistical claims were supported by permutation tests with family-wise error control, which reproduced the same qualitative significance pattern when the signs of correlation coefficients were randomly flipped. Nonetheless, modelling limitations should be acknowledged regarding the robustness of our findings. Every cortical area is currently represented by the same canonical three-population Jansen–Rit NMM. Richer region-specific NMMs that better capture local resting spectra (e.g., with distinct PV and SST interneuron pools) can be swapped in without altering the inference method. Thalamic drivers and finer synaptic-time-constant heterogeneity are omitted. Because the framework is fully generative and modular, these extensions can be incorporated in future work to confirm that the principal conclusions are insensitive to such biophysical detail while giving an even tighter account of regional dynamics.”

3. the authors should emphasize why the use of the model is instrumental in obtaining their results, i.e. why those results could not have been obtained through other means (e.g., Granger causality analyses).

My response: Our conclusions rest on parameters and latent variables—such as region-specific synaptic gains and the system-level Jacobian that quantifies dynamical stability—that are not observable in the data and therefore cannot be estimated with purely statistical tools like Granger-causality (GC) analysis. GC tests whether past activity in one signal improves the linear prediction of another; it neither embodies the nonlinear biophysics that shape cortical rhythms nor distinguishes hidden excitatory and inhibitory drives, and its inferences are well known to be sensitive to filtering, noise, and volume-conduction artefacts [14-16]. By contrast, our neural-mass model is a forward generative description of circuit mechanisms: it reproduces the empirical MEG spectra while simultaneously yielding interpretable parameters whose variations we can relate to changes in conscious state. Previous work comparing generative modelling and GC has shown that only the former can recover the hidden neurophysiological causes of observed activity and predict the effects of perturbations—capabilities that have been exploited in recent whole-brain studies of sleep, anaesthesia and disorders of consciousness [17, 18]. Thus, the model is not merely a convenience: it is the instrument that allows us to uncover mechanisms, test

counterfactual manipulations, and generalise across conditions—results that would remain inaccessible with GC alone.

Regarding this issue, we have added a short discussion in the first paragraph of the Discussion section.

“Crucially, the forward generative nature of the model is itself instrumental. It yields latent neurophysiological variables—region-specific synaptic gains and the eigenvalues of the system Jacobian that cannot be obtained with purely descriptive connectivity measures. Techniques such as Granger causality test linear predictability between observed time-series but neither embody the nonlinear circuit dynamics nor avoid the sensitivity to filtering and volume conduction artefacts that cloud mechanistic interpretation [52]. By fitting our model to reproduce the empirical MEG spectra, we can recover those hidden neurophysiological parameters and state variables allowing for relating neurophysiological variables to the change of consciousness levels. The model is therefore not a mere convenience but the analytical lens through which the stability shifts associated with conscious-state transitions become visible.”

4. is the use of the number of unstable eigenmodes to assess cortical stability standard? If yes, appropriate references should be cited, and if not, this should be justified further, and why other potential metrics could not be used.

My response: Assessing cortical stability via the number of Jacobian eigenmodes whose real part is positive is a standard form of linear-stability analysis in large-scale brain modelling. In one study the authors fitted patient-specific neural-mass networks to intracranial EEG and then evaluated the stability of the system by counting how many Jacobian eigenvalues acquired a positive real part as seizure activity spread [19]. In another study, authors fitted first-order autoregressive models to electrocorticography data (covering the cerebral cortex) and showed that anaesthesia reduced the fraction of positive eigenvalues, indicating a move toward a more stable regime [20]. In both contexts the method serves the same purpose: each positive eigenvalue defines an independent direction in the system’s state-space along which infinitesimal perturbations grow exponentially, so taking the fraction of these modes provides a scale-free, analytically tractable index of how many avenues of instability the cortical network affords. Because it is derived directly from the fitted biophysical parameters—without long, noise-sensitive simulations—and summarises the dimensionality of the unstable manifold rather than just its most unstable direction, this metric offers a parsimonious and mechanistically transparent gauge of cortical stability that is well suited to comparing conditions such as wakefulness, anaesthesia, and pathological excitation.

Regarding this issue, we added a single paragraph in the Discussion section to show the rationale of using this method.

“Evaluating cortical stability via the number of positive eigenvalues is an increasingly common approach in the linear stability analysis in brain modelling [62, 63]. Because every positive eigenvalue marks an independent phase space direction in which infinitesimal perturbations grow, the number or proportion of such modes provides a scale-free, analytically tractable index of how many “routes to instability” the cortex affords, derived directly from fitted biophysical parameters, without lengthy

simulations, and thus offers an interpretable and useful gauge for comparing stability across wakefulness, anaesthesia and pathology.”

5. the Discussion could be fleshed out a bit more, by discussing for example previous works using comparable whole-brain model approaches, and how the present model and associated analyses stands out.

My response: Thank you for pointing out the need to situate our work within the broader literature. We have added a dedicated paragraph in the Discussion section to contrast our framework with the main whole-brain modelling methods. See below.

“Several frameworks already linked whole-brain models to neuroimaging data. The Virtual Brain (VB) fits neural mass equations onto an individual’s diffusion-MRI connectome to reproduce empirical EEG/MEG signatures. In its canonical form the anatomical weights are fixed, and only a handful of global control variables are tuned until simulated spectra and static functional-connectivity match the data [77]. Dynamic mean-field (DMF) formulations compress each region to an excitation inhibition pair whose collective firing rate is then coupled through the empirical connectome, and a single global coupling constant is iteratively adjusted until the model reproduces time-averaged fMRI correlations [78]. Further simplification leads to Hopf/Stuart–Landau oscillators poised near a super-critical bifurcation, which explain resting-state “turbulence” and state-dependent spectral shifts with only a regional bifurcation parameter and a global coupling gain [68]. A variant is the Kuramoto phase model where each area contributes a single phase variable, time delays approximate axonal conduction, and large-scale synchrony emerges from sinusoidal coupling, successfully recapitulating slow BOLD co-fluctuations at minimal computational cost [11]. These frameworks illuminate universal dynamical principles but ignore distinct synapse classes, assume the structural connectome is immutable, and usually infer no more than one or two global parameters. Our framework inherits VB’s commitment to anatomical realism while addressing the above limitations. Each cortical node is a biologically plausible Jansen–Rit neural mass that retains separate pyramidal, excitatory and inhibitory populations. Directly from the empirical MEG timeseries, we estimate time-resolved local synaptic gains, afferent drives and inter-regional coupling strengths. Consequently, the local and inter-regional connection strengths are allowed to drift with conscious states. By recovering multiple neurophysiological variables per region, the framework allows correlating neurophysiological variables to the change of brain conditions in the time-domain and identifies specific brain regions that are responsible for the change. These kinds of inference analyses have not yet been studied with the VB and oscillator-based model frameworks.”

We appreciate the opportunity to revise our manuscript and trust that our responses address the reviewers’ concerns. Thank you for your time and consideration.”

Best regards,
Yun Zhao, PhD on behalf of all authors

- [1] J. D. Sitt, J.-R. King, I. El Karoui, B. Rohaut, F. Faugeras, A. Gramfort, L. Cohen, M. Sigman, S. Dehaene, and L. Naccache, "Large scale screening of neural signatures of consciousness in patients in a vegetative or minimally conscious state," *Brain*, vol. 137, no. 8, pp. 2258-2270, 2014.
- [2] J.-R. King, J. D. Sitt, F. Faugeras, B. Rohaut, I. El Karoui, L. Cohen, L. Naccache, and S. Dehaene, "Information sharing in the brain indexes consciousness in noncommunicative patients," *Current biology*, vol. 23, no. 19, pp. 1914-1919, 2013.
- [3] Y. Zhao, M. Boley, A. Pelentritou, P. J. Karoly, D. R. Freestone, Y. Liu, S. Muthukumaraswamy, W. Woods, D. Liley, and L. Kuhlmann, "Space-time resolved inference-based neurophysiological process imaging: Application to resting-state alpha rhythm," *NeuroImage*, vol. 263, pp. 119592, 2022.
- [4] P. J. Karoly, L. Kuhlmann, D. Soudry, D. B. Grayden, M. J. Cook, and D. R. Freestone, "Seizure pathways: A model-based investigation," *PLoS computational biology*, vol. 14, no. 10, pp. e1006403, 2018.
- [5] Y. Zhao, D. B. Grayden, M. Boley, Y. Liu, P. J. Karoly, M. J. Cook, and L. Kuhlmann, "Cortical stability and chaos during focal seizures: insights from inference-based modeling," *Journal of Neural Engineering*, 2025.
- [6] L. Kuhlmann, D. R. Freestone, J. H. Manton, B. Heyse, H. E. Vereecke, T. Lipping, M. M. Struys, and D. T. Liley, "Neural mass model-based tracking of anesthetic brain states," *NeuroImage*, vol. 133, pp. 438-456, 2016.
- [7] B. H. Jansen, and V. G. Rit, "Electroencephalogram and visual evoked potential generation in a mathematical model of coupled cortical columns," *Biological cybernetics*, vol. 73, no. 4, pp. 357-366, 1995.
- [8] A. Spiegler, S. J. Kiebel, F. M. Atay, and T. R. Knösche, "Bifurcation analysis of neural mass models: Impact of extrinsic inputs and dendritic time constants," *NeuroImage*, vol. 52, no. 3, pp. 1041-1058, 2010.
- [9] O. David, and K. J. Friston, "A neural mass model for MEG/EEG:: coupling and neuronal dynamics," *NeuroImage*, vol. 20, no. 3, pp. 1743-1755, 2003.
- [10] M. Schellenberger Costa, A. Weigenand, H.-V. V. Ngo, L. Marshall, J. Born, T. Martinetz, and J. C. Claussen, "A thalamocortical neural mass model of the EEG during NREM sleep and its response to auditory stimulation," *PLoS computational biology*, vol. 12, no. 9, pp. e1005022, 2016.
- [11] M. S. Costa, J. Born, J. C. Claussen, and T. Martinetz, "Modeling the effect of sleep regulation on a neural mass model," *Journal of computational neuroscience*, vol. 41, pp. 15-28, 2016.
- [12] R. Moran, D. A. Pinotsis, and K. Friston, "Neural masses and fields in dynamic causal modeling," *Frontiers in computational neuroscience*, vol. 7, pp. 57, 2013.
- [13] H. Weigt, K. Föhr, M. Georgieff, E. Georgieff, U. Senftleben, and O. Adolph, "Xenon blocks AMPA and NMDA receptor channels by different mechanisms," *Acta neurobiologicae experimentalis*, vol. 69, no. 4, pp. 429-440, 2009.
- [14] A. K. Seth, A. B. Barrett, and L. Barnett, "Granger causality analysis in neuroscience and neuroimaging," *Journal of Neuroscience*, vol. 35, no. 8, pp. 3293-3297, 2015.
- [15] K. J. Friston, A. M. Bastos, A. Oswal, B. Van Wijk, C. Richter, and V. Litvak, "Granger causality revisited," *Neuroimage*, vol. 101, pp. 796-808, 2014.
- [16] S. Hu, and H. Liang, "Causality analysis of neural connectivity: New tool and limitations of spectral Granger causality," *Neurocomputing*, vol. 76, no. 1, pp. 44-47, 2012.
- [17] I. Mindlin, R. Herzog, L. Belloli, D. Manasova, M. Monge-Asensio, J. Vohryzek, A. Escrichs, N. Alnagger, P. Núñez, and M. Kringelbach, "Whole-brain modelling supports the use of serotonergic psychedelics for the treatment of disorders of consciousness," *bioRxiv*, pp. 2023.12.29.573603, 2023.
- [18] A. I. Luppi, J. Cabral, R. Cofre, P. A. Mediano, F. E. Rosas, A. Y. Qureshi, A. Kuceyeski, E. Tagliazucchi, F. Raimondo, and G. Deco, "Computational modelling in disorders of consciousness: Closing the gap towards personalised models for restoring consciousness," *NeuroImage*, vol. 275, pp. 120162, 2023.
- [19] S. Olmi, S. Petkoski, M. Guye, F. Bartolomei, and V. Jirsa, "Controlling seizure propagation in large-scale brain networks," *PLoS computational biology*, vol. 15, no. 2, pp. e1006805, 2019.
- [20] G. Solovey, L. M. Alonso, T. Yanagawa, N. Fujii, M. O. Magnasco, G. A. Cecchi, and A. Proekt, "Loss of consciousness is associated with stabilization of cortical activity," *Journal of Neuroscience*, vol. 35, no. 30, pp. 10866-10877, 2015.